# Analysis of the Differences in Rhizosphere Microbial Communities and Pathogen Adaptability in Chili Root Rot Disease Between Continuous Cropping and Rotation Cropping Systems

**DOI:** 10.3390/microorganisms13081806

**Published:** 2025-08-01

**Authors:** Qiuyue Zhao, Xiaolei Cao, Lu Zhang, Xin Hu, Xiaojian Zeng, Yingming Wei, Dongbin Zhang, Xin Xiao, Hui Xi, Sifeng Zhao

**Affiliations:** 1Key Laboratory of Oasis Agricultural Pest Management and Plant Protection Resources Utilization, Xinjiang Uygur Autonomous Region, Shihezi University, Shihezi 832003, China; zhaoqiuyue1@stu.shzu.edu.cn (Q.Z.); tulanduocxl@sina.com (X.C.); 13029617900@163.com (L.Z.); huxin@stu.shzu.edu.cn (X.H.); cengxiaojian@stu.shzu.edu.cn (X.Z.); weiyingming@stu.shzu.edu.cn (Y.W.); 20221012355@stu.shzu.edu.cn (D.Z.); 20221012416@stu.shzu.edu.cn (X.X.); 2Xinjiang Production and Construction Corps Key Laboratory of Special Fruits and Vegetables Cultivation Physiology and Germplasm Resources Utilization, Shihezi University, Shihezi 832003, China

**Keywords:** chili root rot, *Fusarium solani*, crop rotation, rhizosphere microbiome, biological characteristics

## Abstract

In chili cultivation, obstacles to continuous cropping significantly compromise crop yield and soil health, whereas crop rotation can enhance the microbial environment of the soil and reduce disease incidence. However, its effects on the diversity of rhizosphere soil microbial communities are not clear. In this study, we analyzed the composition and characteristics of rhizosphere soil microbial communities under chili continuous cropping (CC) and chili–cotton crop rotation (CR) using high-throughput sequencing technology. CR treatment reduced the alpha diversity indices (including Chao1, Observed_species, and Shannon index) of bacterial communities and had less of an effect on fungal community diversity. Principal component analysis (PCA) revealed distinct compositional differences in bacterial and fungal communities between the treatments. Compared with CC, CR treatment has altered the structure of the soil microbial community. In terms of bacterial communities, the relative abundance of Firmicutes increased from 12.89% to 17.97%, while the Proteobacteria increased by 6.8%. At the genus level, CR treatment significantly enriched beneficial genera such as *RB41* (8.19%), *Lactobacillus* (4.56%), and *Bacillus* (1.50%) (*p* < 0.05). In contrast, the relative abundances of *Alternaria* and *Fusarium* in the fungal community decreased by 6.62% and 5.34%, respectively (*p* < 0.05). Venn diagrams and linear discriminant effect size analysis (LEfSe) further indicated that CR facilitated the enrichment of beneficial bacteria, such as *Bacillus*, whereas CC favored enrichment of pathogens, such as *Firmicutes*. *Fusarium solani* MG6 and *F. oxysporum* LG2 are the primary chili root-rot pathogens. Optimal growth occurs at 25 °C, pH 6: after 5 days, MG6 colonies reach 6.42 ± 0.04 cm, and LG2 5.33 ± 0.02 cm, peaking in sporulation (*p* < 0.05). In addition, there are significant differences in the utilization spectra of carbon and nitrogen sources between the two strains of fungi, suggesting their different ecological adaptability. Integrated analyses revealed that CR enhanced soil health and reduced the root rot incidence by optimizing the structure of soil microbial communities, increasing the proportion of beneficial bacteria, and suppressing pathogens, providing a scientific basis for microbial-based soil management strategies in chili cultivation.

## 1. Introduction

Chili (*Capsicum annuum* L.), a member of the Solanaceae family, is widely used as a vegetable, spice, natural coloring, and traditional medicine and has significant economic value [1,2,3]. In recent years, China′s chili planting area has stabilized at more than 73.3 × 10^4^ ha, and its production ranks first in the world. It has an extensive cultivation area, covering Xinjiang, Guizhou, Henan, and other provinces [4]. Located in northwest China, Xinjiang has a temperate continental climate with abundant light and dry air, which is an ideal environment for growing chili. Xinjiang has become an important raw material base for chili in the country, with a planting area of 100,800 hectares and an output of up to 4,131,000 tons in 2021, making it one of the most important local specialty crops [5].

Intensive and continuous chili cultivation has been widely adopted in Xinjiang. However, with increasing market demand and economic incentives, the barriers to continuous cropping have gradually become a common challenge to chili cultivation [6]. Negative soil effects caused by continuous cropping include nutrient imbalance, chemosensor accumulation, soil physicochemical property degradation, microbial community structure alteration, and plant autotoxicity [7]. The combination of these factors leads to slow plant growth, reduced yield and quality, and increased disease incidence, even under good field management conditions [8]. Long periods of continuous cultivation (more than 10 years) have been shown to reduce chili yield by up to 70% and significantly affect the quality [9]. Crops, such as tobacco, licorice, Codonopsis, and groundnut, are susceptible to soil-borne diseases under continuous cropping conditions, resulting in severe economic losses [10,11,12,13]. Therefore, addressing continuous cropping barriers in chili is essential to ensure sustainable production.

With the increase in the duration of continuous cultivation, the occurrence of soil-borne diseases in chili cultivation has become more prominent, limiting production [14]. Soil-borne diseases refer to those in which pathogens are latent in the soil and infect the crop under suitable conditions, and their occurrence is closely linked to continuous cropping [15]. Chili is highly susceptible to soil-borne diseases, especially chili root rot, which affects yield and quality [16]. Chili root rot is mainly caused by *Fusarium*, *Pythium*, and *Phytophthora* [17]. These pathogens enter through micro-wounds in the roots and stems. At disease onset, the plant wilts during the day and recovers in the morning and evening, as the underground fibrous roots are constricted and rotted, and the rootstock becomes waterlogged and necrotic. The cortex becomes brown and easily detached, and the vascular bundles turn brown. Finally, the rootstock rots, and the plant dies [18]. Pathogens survive winter as mycelium or chlamydospores in the soil. They spread via rain splash, irrigation water, or infected seed, thriving under high humidity, wide day–night temperature swings, and continuous monocropping [19]. At present, chili root rot is primarily controlled by chemical pesticides, but long-term use leaves behind pesticide residues and increases resistance to pathogens. Therefore, green and sustainable control methods, such as biological control and soil improvement, require further development [20].

The emergence of soil microbiology has provided new perspectives on plant disease prevention and control. The rhizosphere, as the core of plant water and nutrient uptake, microbial activity, and plant–soil–microbe interactions, is a key site for microbial influence on plant health [21]. Rhizosphere microbiomes promote nutrient uptake and use by plants through mineralization, adsorption, and fixation of nutrients, regulation of organic matter decomposition, and carbon and nitrogen cycling [22]. In addition, plant-associated microbiomes can inhibit pathogen invasion and reduce the incidence of soil-borne diseases [23]. Studies have shown that root-associated microorganisms significantly influence plant health by enhancing plant stress tolerance, participating in biochemical cycles, and regulating plant growth and community composition [24]. Beneficial microorganisms refer to functional microbial groups that establish mutualistic relationships with plants and enhance plant growth/stress resistance through multiple mechanisms, primarily including plant growth-promoting rhizobacteria (PGPR), endophytic bacteria (EB), and arbuscular mycorrhizal fungi (AMF) [25]. Beneficial microorganisms, such as *Trichoderma* and *Pseudomonas*, have shown great potential in disease suppression [26]. For example, the dominant rhizosphere flora of healthy tomato significantly reduced the disease incidence, demonstrating the potential of its microbiome in disease prevention and control [27]. Purposefully adjusting the structure of the rhizosphere microbial community in crops and actively introducing beneficial microorganisms can significantly reduce disease incidence and enhance the disease resistance of crops [28]. Related studies have demonstrated that synthetic microbial communities (SynComs) constructed with different microbial proportions exhibit remarkable effects in plant disease management and improving crop disease resistance, providing new perspectives for the innovation and development of disease prevention and control strategies. Exogenous application of SynComs has been shown to significantly enhance plant disease resistance, as evidenced in the control of peanut root rot [29] and Astragalus root rot [30]. In addition, the nutritive organs (roots and stems) of chili have been shown to influence soil microbial community composition significantly more than the reproductive organs (fruit), and its roots recruited beneficial microbes to defend against pathogen invasion [31]. Glucanase in *Trichoderma* has been closely associated with root rot development in chili [32]. Optimizing the soil environment for *Pseudostellaria heterophylla* and modulating the light conditions for *Panax notoginseng* growth substantially alters the composition of the soil microbial community, thereby effectively suppressing the root rot incidence [33,34]. These results indicate that soil microorganisms play an important role in plant disease prevention and control and provide a theoretical basis for the development of green and sustainable disease management strategies.

Crop rotation is an important strategy for reducing crop barriers and reducing the incidence of soil-borne diseases [13]. Crop rotation activates different microbial populations, including beneficial microorganisms that promote plant growth [3]. For example, crop rotation has been shown to significantly reduce the root rot incidence in groundnut, as the rhizosphere microbiota effectively inhibited the growth of pathogenic bacteria [29]. In addition, rotation of oilseed rape with peas and barley not only increased the yield and oil content but also significantly altered the structural composition of the soil fungal community [35]. Pineapple–banana rotation was more effective in reducing soil organic carbon and suppressing banana wilt than maize–banana rotations. In both rotation systems, fungal communities were more variable than bacterial communities. Specifically, the pineapple–banana rotation significantly increased the abundance of Acidobacteria, Planctomycetes, Chloroflexi, Gp1, Gp2, and *Burkholderia* but decreased the abundance of Basidiomycota [36]. These changes in microbial communities may be key factors in reducing the incidence of wilt. Globally, crop rotation has long been recognized as a critical strategy in suppressing soil-borne pathogens and promoting beneficial microbial populations. In the U.S., corn–soybean rotations enhanced rhizosphere *Pseudomonas* and *Bacillus* populations, which suppressed *Fusarium* and *Rhizoctonia* diseases [37]. Similarly, In European and Mediterranean cropping systems, rotating wheat with tomato effectively suppresses Fusarium wilt. This rotation strategy enhances tomato performance by increasing the abundance of antagonistic microorganisms, optimizing soil chemical properties, and restructuring the soil ecological network [38]. Despite previous evidence that crop rotation reshapes soil microbial communities, how rotation suppresses root rot is still unclear. Moreover, investigations of the specific microbial functional groups that are activated after rotation and how these microorganisms interact with soil-borne pathogens are still lacking in systematic research. Therefore, we compared chili rhizosphere microbes under continuous cropping and rotation. We then tracked how rotation enriches beneficial taxa and reduces root-rot pathogens, aiming to support green disease management.

In this study, we systematically analyzed the diversity and composition of soil microbial communities in the rhizosphere of diseased chili under continuous and rotational cropping conditions and determined their relationship with the occurrence of root rot. The structural characteristics of soil bacterial and fungal communities were comprehensively analyzed under the two cropping modes, and the effects of different treatments on the rhizosphere microbial communities were assessed. The results of this study contribute to an in-depth understanding of the response mechanisms of the rhizosphere microbial community of chili to changes in cropping patterns and provide a scientific basis for disease control strategies based on soil microbial regulation.

## 2. Materials and Methods

### 2.1. Soil Sample Collection and Processing

In the primary chili cultivation region of Anjihai Town, Shawan City, Tacheng Prefecture, Xinjiang Uygur Autonomous Region, China, for comparative analysis, two distinct cropping systems were examined: a 10-year continuous chili cropping field (85°47′10″ E, 44°37′12″ N) and a cotton–chili rotation field with annual cycles (85°47′39” E, 44°37′31″ N). The experimental sites were located 720 m apart with minimal elevation difference (<3 m) and shared identical soil properties (grey desert soil, pH 8.1 ± 0.4). From each system, plants showing characteristic disease symptoms and their rhizosphere soils were collected for subsequent pathological and microbiological characterization. For chili plants exhibiting root rot symptoms, we implemented a five-point sampling method: five diseased plants were collected at each of five sampling points per field, yielding 25 samples per field (50 total samples from both fields). All samples were properly labeled for subsequent pathogen isolation. At each sampling point, three rhizosphere soil samples from diseased chili plants were simultaneously collected. Rhizosphere soil collection involved removing the top 5 cm of soil around the diseased plants, uprooting the entire plant, and shaking off loosely adhered soil from the roots. A sterile brush was used to gently collect the remaining soil attached to the roots, which was placed in a 10 mL sterile centrifuge tube and promptly transported to the laboratory for storage at −80 °C [39]. All samples were processed immediately after collection to preserve their integrity and viability.

### 2.2. Microbial Sequencing and Sequence Analysis

Samples were sent to Beijing Novogene Technology Co., Ltd. (Beijing, China) to perform total DNA extraction, PCR amplification, and library construction using the NovaSeq 6000 platform for high-throughput sequencing. For data analysis, sample reads were demultiplexed based on barcode and primer sequences. FLASH (V1.2.7) was used to merge paired-end reads into raw tags. Low-quality fragments and short tags were removed through quality control using QIIME 2 [40], resulting in clean tags. Chimeric sequences were then identified and removed using VSEARCH to obtain effective tags. *Operational taxonomic unit* (OTU) clustering was performed at a 97% sequence similarity threshold using Uparse (V7.0.1001) [41], with the most abundant sequence selected as representative for each OTU. Species annotation was conducted by comparing representative OTU sequences against the SILVA 138 database, followed by a statistical analysis of community composition. Phylogenetic relationships were constructed via multiple sequence alignment of representative OTUs using MUSCLE (V3.8.31). The sample data were normalized to facilitate subsequent alpha and beta diversity analyses.

### 2.3. Pathogen Isolation and Identification

To isolate pathogens, the plant roots and stems were washed with clean water to remove impurities and then rinsed under running water for 2 h [42]. On a clean bench, they were then disinfected with 70% ethanol for 1 min and 2% sodium hypochlorite for 6 min and rinsed three times with sterile water. Sterilized scissors were used to cut the roots and stems of the diseased plants into 0.5 × 0.5 cm pieces [43]. After absorbing the excess water, tissue blocks were used to inoculate potato dextrose agar (PDA) plates, which were then incubated at 25 °C in the dark for 3 days, and the number of colonies was counted. Subsequently, mycelia from the edge of each colony were transferred for individual culture. After 5 days, single-spore isolation was performed. Briefly, the mycelia from the edge of the colony were placed into a 1.5 mL centrifuge tube with 1 mL of sterile water and shaken to disperse the spores. This was diluted to a concentration of 10^−3^, and an inoculation loop was used to collect a small amount of the diluted solution, which was then streaked onto a PDA plate. The plates were incubated for 48 h, and single colonies were transferred to new PDA plates. The plates were incubate at 25 °C for 5–7 days to obtain pure cultures.

Based on the colony′s morphological characteristics, 20 typical strains were selected for pathogenicity determination. A small amount of purified mycelium was used to inoculate potato dextrose broth (PDB) liquid medium and cultured at 28 °C with shaking at 180 rpm for 5 days. After cultivation, the spore suspension was collected by filtering through double-layer sterile gauze and diluted to a concentration of 1 × 10^7^ CFU/mL for later use. Chili ‘NMCA10399′ seeds of uniform size and with plump grains were selected. The seeds were first soaked in distilled water at 55 °C for 10 min, followed by immersion in 5% sodium hypochlorite solution for 10 min, and then rinsed three times with sterile water. The treated seeds were placed in 250 mL conical flasks and incubated on a shaker at 180 rpm for 5−7 days at 28 °C until germination. Germinated seeds were transferred to germination boxes lined with sterile filter paper moistened with water and cultured at 25 °C in the dark for 5 days. When the seedlings′ stems reached 2−3 cm in length, they were transplanted into 20 × 15 cm hydroponic boxes and secured with absorbent cotton. Once the seedlings developed 4 true leaves, 500 mL of the 1 × 10^7^ CFU/mL isolate suspension was used to inoculate each plant, and they were co-cultured at 25 °C for 24 h. After inoculation, the plants were transferred to Hoagland nutrient solution for continued growth. Disease progression was monitored every 7 days. The experiment was conducted with 3 replicates, with 12 chili plants per hydroponic box.

For the highly pathogenic strains, the genomic DNA of the fungi was extracted using a Biospin Fungus Genomic DNA Extraction Kit (Hangzhou Bioer Technology Co., Ltd., Hangzhou, China) and stored at −20 °C for future use. PCR amplification was conducted using the universal fungal primers ITS1 (5′-TCCGTAGGTGAACCTGCGG-3′) and ITS4 (5′-TCCTCCGCTTATTGATATGC-3′) as well as the specific primers EF-1H (5′-ATGGGTAAGGAAGACAAGAC-3′) and EF-2T (5′-GGAAGTACCAGTGACATGTT-3′) [44,45]. After the PCR products were detected using 1% agarose gel electrophoresis, they were sent to Sangon Biotech (Shanghai, China) Co., Ltd. for Sanger sequencing. The sequencing results were compared with the NCBI database using BLAST (https://blast.ncbi.nlm.nih.gov/Blast.cgi, accessed on 11 June 2025) to identify the isolates to species. Reference sequences with a high sequence identity to the isolates were selected, and a phylogenetic tree was constructed using Mega 7.0 software to determine their taxonomic status. For the morphological identification of the strains, PDA plates were inoculated with the isolates and cultured. After 3 days, a sterile cover slip was inserted at an angle into the edge of the colony and cultured for another 3 days. The cover slip was then removed, observed under an optical microscope, and photographed, and the morphological characteristics of the conidiophores were recorded. A small amount of mycelium was used to inoculate PDB liquid medium and cultured on a shaker at 180 rpm for 5 days at 25 °C. The spore suspension was collected by filtering through double-layer gauze, and conidia morphology was observed under a microscope, with their size measured and recorded [46].

### 2.4. Determination of the Biological Characteristics of Pathogens

This study experimentally investigated the effects of temperature, pH, carbon source, and nitrogen source on the mycelial growth and spore production of pathogenic fungi. PDA medium was inoculated with mycelial discs and cultured for 7 days. Then, 0.5 cm diameter discs were cut using a punch and placed in biochemical incubators at 5, 15, 20, 25, 30, 35, 40 °C for 5 days. Colony diameters were measured. After 10 days, the discs were crushed in sterile water, and the spore counts were determined using a hemocytometer. The pH of the PDA medium was adjusted to 4, 5, 6, 7, 8, 9, and 10. After inoculating the discs and culturing them at 25 °C for 5 days, the colony diameters and spore counts were measured. In the carbon source experiment, Czapek′s agar medium was used as the base, and glucose, sucrose, soluble starch, lactose, mannitol, maltose, D-xylose, D-fructose, and inositol were added, with no carbon source added as the control [47]. The cultures were incubated at 25 °C, and the colony diameters and spore counts were measured after 5 and 10 days. In the nitrogen source experiment, Czapek′s agar medium was used as the base, and ammonium nitrate, sodium nitrate, potassium nitrate, peptone, yeast extract, beef extract, glycine, L-leucine, and urea were added, with no nitrogen source added as the control. The cultures were incubated at 25 °C, and the colony diameters and spore counts were measured after 5 and 10 days. All experiments were conducted with three replicates.

### 2.5. Statistical Analysis

Multiple bioinformatics tools were employed to analyze the microbial community data. Qiime (Version 1.9.1) was used to calculate the alpha diversity indices, including observed OTUs, Chao1, Shannon, Simpson, ACE, Good′s Coverage, and PD Whole Tree. Inter-group difference analysis was conducted in R software (Version 4.3.0) using *T*-tests and Wilcoxon rank sum tests. Prior to parametric analysis, normality (Shapiro–Wilk test) and homogeneity of variance (Levene test) were performed. When both assumptions were met, an independent-samples *t*-test was used [48]. The unweighted pair–group method with arithmetic mean (UPGMA) clustering tree was constructed based on the Unifrac distance, and PCA was performed using R packages ‘ade4′ and ‘ggplot2′. Linear discriminant analysis (LDA) effect size (LEfSe) software (https://www.bioincloud.tech, 11 June 2025) was utilized for effect size analysis with an LDA score threshold of 4. Metastats analysis was conducted at each taxonomic level (from phylum to species) using R software (Version 4.3.0), with *p*-values calculated using permutation tests and corrected to q-values with the Benjamini–Hochberg method. Analysis of similarities (ANOSIM), multi-response permutation procedure (MRPP), and Adonis analyses were completed using functions in the R ‘vegan′ package (‘anosim′, ‘mrpp′, and ‘adonis′, respectively), and analysis of molecular variance (AMOVA) was implemented using mothur software. Significant differences in species abundance were analyzed using *T*-tests and visualized in R software. Other experimental data were statistically analyzed using IBM SPSS Statistics 23 (Version R23.0.0.0), and graphs were created with OriginPro 2024.

## 3. Results

### 3.1. Diversity Analysis of Rhizosphere Soil Microbial Communities in the Continuous and Rotational Cropping of Chili

In this study, the rhizosphere soil microbial community of diseased chili was systematically analyzed under two cropping patterns: continuous cropping (CC) and crop rotation (CR). Alpha diversity analysis revealed a significantly lower Shannon index in CR bacterial communities compared to CC (*p* < 0.05), though other indices (Chao1 and Observed species) showed no significant differences between treatments. In contrast, fungal communities exhibited no significant differences in any alpha diversity metrics (Chao1, Observed species, or Shannon indices) between CR and CC treatments, indicating minimal rotation effects on fungal diversity. Further PCA based on the Bray–Curtis distance algorithm showed that the rhizosphere soil bacterial communities were separated under the two cropping modes, with the first and second principal components explaining 24.22 and 20.50% of the variance, respectively (Figure 1G). The PCA results also indicated that the fungal communities were farther apart under the two modes, explaining 28.91 and 20.89% of the variance (Figure 1H). These results indicate that CR significantly altered the diversity and community composition of the soil bacterial community in the rhizosphere of chili compared to CC, but the effect on the fungal community was limited.

### 3.2. Relative Abundance Analysis of Dominant Groups in Rhizosphere Soil Microbial Communities Under Two Cropping Patterns

To reveal the effects of CC and CR cropping patterns on the rhizosphere soil microbial community, the relative abundance of the dominant groups in the microbial community was analyzed under both patterns. At the phylum level, the dominant bacterial phyla included Firmicutes, Acidobacteriota, Actinobacteriota, and Proteobacteria (Figure 2A), among which CR increased the relative abundance of Firmicutes and Proteobacteria by 5.07% and 6.80% compared to CC. At the genus level, the dominant genera were *RB41*, *Lactobacillus*, *Bacillus*, *Dubosiella*, and *Romboutsia*, and CR increased the relative abundance of *RB41*, *Lactobacillus*, and *Bacillus* compared to CC (Figure 2B). In the fungal community, the dominant phyla were Ascomycota, Basidiomycota, Mortierellomycota, Chytridiomycota, and Oloidiomycota. Furthermore, compared with CC, CR reduced the relative abundances of the Ascomycota and Basidiomycota phyla by 17.51% and 0.76% respectively (Figure 2C). Genus-level analysis (Figure 2D) showed that *Cephalotrichum*, *Alternaria*, *Fusarium*, *Myrothecium*, and *Chaetomium* were the dominant genera, and CR reduced the relative abundance of *Cephalotrichum*, *Alternaria*, and *Fusarium* (by 1.80%, 6.62%, and 5.34%, respectively). In addition, heat map analysis showed that the relative abundance of *Bacillus*, *Alistipes*, *RB41*, *UTCFX1*, and *Lactococcus* was higher in CR than in CC (Figure 2E), whereas CR treatment reduced the relative abundance of *Cephalotrichum*, *Fusarium*, and *Gibberella* in the soil (Figure 2F). These results indicate that crop rotation significantly altered the abundance and composition of bacterial and fungal communities in rhizosphere soil.

### 3.3. Analysis of Rhizosphere Microbial Community Differences in Chili Under Continuous and Rotation Cropping

Venn diagram analysis showed that the number of CC-specific bacterial OTUs was 851, which was slightly higher than that of CR, with 719 OTUs. The number of OTUs shared by both was 3977 (Figure 3A). In the fungal community, the number of CC-specific OTUs was 181, and the number of CR-specific OTUs was 178. The number of shared OTUs was 186 (Figure 3B). Although there were some differences in bacterial and fungal community composition between the CC and CR treatment groups, these differences were relatively small overall. To further explore the effects of CC and CR treatments on the soil microbial community, we conducted a genus-level intergroup *T*-test. In the bacterial community, CR significantly increased the relative abundance of *Bacillus*, *AKYG587*, *unidentified_Chloroplast*, *Lysinibacillus*, *Luteibacter*, *TM7a*, and *Salinimicrobium* (Figure 3C), but in the fungal community, CR significantly reduced the relative abundance of *Fusarium*, *Gibberella*, *Didymella*, *Sordariomycetes*_sp, and *Pleospora* (Figure 3D).

To explore the effects of CC and CR treatments on soil microbial communities in depth, we performed LEfSe analyses of both treatments to identify statistically significant biomarkers. In the soil bacterial community, 23 biomarkers were identified in CC, and eight biomarkers were identified in CR (Figure 4A). When the LDA score threshold was set to 4, genus-level biomarkers in CC included *Roseisolibacter*, *Solirubrobacter*, and *Microvirga*, and biomarkers in CR were *RB41* and *Bacillus* (Figure 4B). In the soil fungal community, 12 biomarker species were found in CC, and 10 biomarker species were identified in CR (Figure 4C). The histogram of the distribution of LDA values showed that biomarkers at the genus level in CC included *Alternaria*, *Fusarium*, *Gibberella*, *Pseudombrophila*, and *Cephalotrichum*, and biomarkers in CR included *Cladosporiaceae*, *Lulworthia*, and *Wyrothecium* (Figure 4D). Combining the results of T-test and LEfSe analyses, we hypothesized that the increase in *Bacillus* and decrease in *Fusarium* may be the key factors in crop rotation mitigating the occurrence of root rot. These results suggest that crop rotation positively affects plant health by modulating the composition of soil microbial communities, especially by increasing the relative abundance of beneficial bacteria and inhibiting the growth of potentially pathogenic bacteria.

### 3.4. Isolation and Characterization of Chili Root Rot Pathogens

A total of 132 pathogen strains were isolated from chili under CC and CR treatments in Anjihai Town, Xinjiang: 30 from stems (LJ) and 40 from roots (LG) under CC, and 15 from stems (MJ) and 47 from roots (MG) under CR. A total of 19 isolates were screened based on morphological characteristics (including color, mycelial morphology, pigment production, and spore morphology) of the colony, and their pathogenicity was determined using hydroponic inoculation experiments. The results of the pathogen inoculation experiment showed that there was a significant differentiation in pathogenicity among the tested strains. The dynamic observation after inoculation revealed that four strains, MG6, MG5, LG3, and LG2, could induce typical symptoms of chili root rot in the chili plants: initially, the lower leaves wilted and drooped (5–7 days after inoculation), then developed systemic chlorosis (8–12 days), finally leading to the complete death of the entire plant (14–20 days) (Figure 5A). Disease index statistics showed that the disease indices of these strains were significantly higher than those of the control group and other treatments (*p* < 0.05). Among them, MG6 had the strongest pathogenicity (disease index 85.80 ± 2.23), followed by MG5 (74.69 ± 2.22), LG3 (59.87 ± 2.69), and LG2 (54.93 ± 0.62) (Figure 5B, data are presented as mean ± standard error). It is noteworthy that the initial symptoms appeared 3 days earlier in the MG6 treatment group compared to LG2, and the disease progression rate was also faster. Further molecular characterization was performed by PCR amplification with universal primers ITS1/ITS4 and specific primers EF-1H/EF-2T, which yielded bands of approximately 550 bp (Figure 5C) and 700 bp (Figure 5D), respectively. After sequencing comparison, a phylogenetic tree was constructed based on the ITS and EF sequences. MG5, MG6, and LG3 clustered with *Fusarium solani* (99.87% similarity), and LG2 clustered with *Fusarium oxysporum* (99.51% similarity) (Figure 5E).

Combining the results of pathogenicity determination and molecular characterization, MG6 and LG2 were selected as representative strains for subsequent experiments. Strain MG6 grew rapidly on PDA medium, and the colony diameter reached 4.5–5.5 cm on day 5 (Figure 5(F1)). Its aerial mycelium was thin and fluffy, and its color was white, with yellow pigmentation. Microscopic observation showed that the large conidia of MG6 were matt type with 3–5 septa, measuring 35–50 × 3.7–5 μm (Figure 5(F2)), and the small conidia were kidney type, measuring 8–16 × 2.5–4 μm (Figure 5(F3)), and were pseudo-head-like attached to spore-producing cells (Figure 5(F4)). The spore-producing cells were mostly longer single vial peduncles and were produced in clusters on the conidiophore (Figure 5(F5)). Strain LG2 grew more slowly than MG6 on PDA medium, and the diameter of the colony was 3.7–4.8 cm after 5 days. Its colony surface was covered with white fluffy mycelium, and the center of the colony produced a mauve pigment, accompanied by a faint irritating odor. The mycelium was arranged tightly (Figure 5(G1)). Microscopic observation showed that the large conidia of LG2 were slightly obtuse at both ends and slightly pointed at the top, with a size of 12–15 × 2.6–4.7 μm (Figure 5(G2)); the small conidia were fusiform (Figure 5(G3)), pseudocapitate, and attached to the spore-producing cells (Figure 5(G4)). The spore-producing cells were short and had a single vial peduncle structure (Figure 5(G5)). Based on molecular identification and morphological features, strain MG6 was identified as *Fusarium solani*, and strain LG2 was identified as *Fusarium oxysporum*.

### 3.5. Biological Characterization of Pathogens

To investigate the biological characteristics of strains MG6 and LG2 and clarify their growth and spore production patterns under different environmental conditions, the present study systematically analyzed the growth and spore production ability of the two pathogen strains under different temperatures, pHs, and carbon and nitrogen sources. Strain MG6 had the largest colony diameter (6.12 cm) and the highest spore production (4.07 × 10^6^ CFU/mL) at 25 °C, indicating that this temperature was optimal for its growth and spore production (Figure 6A). At a pH of 6, the colony diameter (6.42 cm) of MG6 reached the maximum, while the spore production (2.7 × 10^6^ CFU/mL) was not significantly higher than that under other pH treatments but showed better growth overall (Figure 6B). In terms of carbon sources, sucrose, mannitol, maltose, and inositol significantly promoted the growth of MG6 mycelium, with colony diameters of up to 5.8 and 5.78 cm when inositol and maltose, respectively, were used as the carbon sources. Spore production was highest with inositol (2.17 × 10^6^ CFU/mL), suggesting that these carbon sources had an important influence on the growth and spore production of MG6 (Figure 6C). In terms of nitrogen sources, potassium nitrate and peptone significantly promoted the growth of MG6 hyphae, while ammonium nitrate showed some inhibitory effects. All nitrogen sources promoted spore production, with the highest spore production (1.88 × 10^6^ CFU/mL) observed under sodium nitrate treatment, but there was no significant difference between beef paste and L-leucine treatments (Figure 6D).

For the LG2 strain, its colony diameter was 4.25 cm at 25 °C, and spore production reached 2.37 × 10^6^ CFU/mL (Figure 6E). At a pH of 6, the diameter of LG2 colonies was the largest (5.33 cm) and showed a trend of increasing and then decreasing with the increase in pH, indicating that a pH of 6 was optimal for its growth (Figure 6F). At a pH of 6 and 7, the spore production was 2.06 × 10^6^ and 1.86 × 10^6^ CFU/mL, respectively, and the overall changes were stable. In terms of carbon source, maltose significantly promoted the mycelial growth of LG2, and the colony diameter reached 5.63 cm. Soluble starch showed a certain inhibitory effect. LG2 almost did not produce spores on the medium without an added carbon source, and the highest spore production was observed on the medium with added D-fructose (1.87 × 10^6^ CFU/mL), indicating that D-fructose was the most suitable carbon source for spore production (Figure 6G). In terms of nitrogen sources, potassium nitrate significantly promoted the growth of LG2 mycelium, with colony diameter up to 5.56 cm, while beef paste significantly promoted spore production, with a spore yield of 1.08 × 10^6^ CFU/mL, indicating that different nitrogen sources had significant regulatory effects on the growth and spore production of LG2. In contrast, ammonium nitrate and urea significantly inhibited spore production (Figure 6H).

## 4. Discussion

In recent years, studies have shown that the soil microbial community structure, nutrient efficiency, and chemosensitivity are key factors contributing to crop succession disorders [49]. Continuous medicinal-plant monocropping increases root-rot incidence to 3.5-fold that of newly reclaimed land [50], underscoring the need to break replant barriers for yield, soil health, and the environment. Rather than indiscriminately raising microbial diversity, crop rotation exerts selective pressure that enriches beneficial taxa and suppresses pathogens, thereby reducing disease risk and stabilizing yields. This targeted re-structuring of the microbiome is accompanied by improved soil nutrient status and lower pesticide dependence [51]. Consistent findings have been reported for chili production: adopting different cultivation modes after chili planting markedly decreased the relative abundance of potentially pathogenic fungi while increasing that of beneficial counterparts, further optimizing microbial community structure and enhancing soil disease resistance and fertility. Similarly, alfalfa–forage rotations reassemble bacterial communities and mitigate replant disorders [52,53]. Thus, the agronomic benefit of rotation stems from its selective pressure, which lowers Shannon evenness while directionally enriching functional taxa and suppressing pathogens, thereby enhancing soil health and crop productivity.

### 4.1. Differences in Alpha/Beta Diversity of Soil Microorganisms in the Rhizosphere of Chili Under Continuous Cropping and Rotation Systems

Crop rotation reshapes the chili rhizosphere microbiome, yet its primary impact lies in altering evenness rather than species richness. Following CR, the bacterial Shannon index declined significantly, whereas the Chao1 and Observed_species indices were slightly lower but not statistically different, indicating that CR restructures the community through selective enrichment of specific functional taxa without genuinely changing the total species pool. Likewise, fungal richness remained stable, while a higher Shannon index pointed to internal re-assembly, consistent with peanut rotation studies reporting decreased bacterial diversity and increased fungal diversity [54,55].

PCA further revealed clear separation of both bacterial and fungal communities between the two cropping pattern, with bacteria being more sensitive to the shift in cultivation pattern [29]. Earlier work showed that continuous potato cropping can elevate bacterial diversity [56], whereas a tobacco–faba bean–lettuce–oilseed rape rotation lowered microbial diversity by modulating soil pH and other physicochemical properties [57]. In summary, a well-designed rotation regime safeguards soil health and boosts crop productivity by precisely modulating community evenness and the abundance of functional taxa.

### 4.2. Crop Rotation Suppresses Root Rot Pathogens by Regulating Microbial Community Composition

Rhizosphere microbes are vital for plant health and productivity [58]. They are also the frontline where soil-borne diseases develop and where microbes, pathogens, and plants interact [59,60]. This study demonstrates that different farming practices significantly alter the structure and function of soil microbial communities, with particularly pronounced differences in bacterial taxa abundance and diversity. Under CR treatment, the relative abundances of two beneficial bacterial phyla, Firmicutes and Proteobacteria, increased by 5.07% (from 12.90% to 17.97%) and 6.8% (from 11.83% to 18.63%), respectively, compared to CC. Importantly, antagonistic bacteria commonly associated with Firmicutes, including *Bacillus* and *Lactobacillus*, showed significant enrichment in CR treatment in our study region. These results suggest that rotation practices may effectively enhance soil disease suppression capacity and ecosystem stability by rebuilding beneficial microbial communities [61]. At the genus level analysis, probiotic groups including *RB41* (8.18%), *Lactobacillus* (4.26%), and *Bacillus* (1.50%) exhibited higher abundances in CR treatment, while potential pathogens such as *Fusarium* showed decreased abundance. These findings further confirm that crop rotation may selectively restructure microbial niches through rhizosphere environment modification, thereby promoting colonization and functional expression of beneficial microorganisms [62].

*RB41* is an important member of Acidobacteria and also a key participant in the soil nitrogen cycle. It regulates soil nitrogen forms and increases pH through nitrate reduction (NO_3_^−^ → NH_4_^+^). This alkaline environment can effectively inhibit the growth of acidophilic pathogens such as *Fusarium*. Studies have found that the abundance of *RB41* is significantly positively correlated with soil pH (optimal pH 8.4), and can be used as a biological indicator of soil ecological health [63,64]. In this study, CR significantly promoted the proliferation of *RB41* by alleviating soil acidification caused by continuous cropping, indicating that the rotation mode can effectively improve the micro-environment of the root zone. *Lactobacillus* and *Bacillus* are functionally important beneficial microorganisms in rhizosphere ecosystems. *Lactobacillus* mediates direct antagonism against pathogenic fungi via secretion of organic acids and antifungal peptides [65]. In parallel, *Bacillus* exerts multifaceted biocontrol activity through biosynthesis of antimicrobial compounds, including antibiotics and siderophores, competitive exclusion of pathogens, and induction of plant systemic resistance [66]. Current research substantiates that *Bacillus* achieves particularly robust biocontrol outcomes through dual mechanisms of antimicrobial production and host resistance induction [67,68]. Meanwhile, CR markedly reduced the relative abundance of Fusarium, a pathogen implicated in a wide range of plant diseases [69,70,71]. The results of this study are consistent with those of previous studies. Corn–soybean rotation can reduce pathogen abundance in the rhizosphere and increase the relative abundance of certain beneficial fungi [72]. Thus, crop rotation can reduce disease incidence by enriching beneficial bacteria and inhibiting pathogenic fungi, maintaining soil ecological functions, and thereby promoting healthy plant growth.

### 4.3. Stability of the Rhizosphere Core Microbiota

Soil health serves as the cornerstone for sustainable agricultural development, with its essence lying in maintaining diverse beneficial microbial communities (including bacteria, fungi, and algae) [73]. These “soil guardians” support agricultural production through three key mechanisms: optimizing nutrient cycling efficiency, suppressing soil-borne pathogens, and enhancing crop yield and quality. Of particular concern is that under soil degradation conditions, the pathogen-suppressing function of beneficial microbial communities is often the first to be compromised. Therefore, establishing a microbial function-based soil health monitoring system has become a crucial measure for ensuring sustainable agricultural development.

Although CR reshaped the microbial community, Venn analysis revealed shared OTUs across both treatments. This suggests a stable core microbiome persists in chili rhizospheres. These core microorganisms may play a key role in maintaining soil health and promoting plant growth [74]. The persistent core taxa comprised functionally complementary microorganisms that have been linked to plant growth promotion in previous studies [75]. Moreover, *t*-tests and LEfSe analyses consistently showed that CR significantly enriched the plant-growth-promoting bacteria *Bacillus*, *Lysinibacillus*, and *TM7a*, while reducing the relative abundances of the pathogenic fungi *Fusarium* and *Gibberella*. These results further confirm that rotation suppresses chili root rot by restructuring the microbial community.

### 4.4. Effect of Crop Rotation on Chili Root Rot Pathogens

This study confirms that the primary pathogens causing chili root rot in Anjihai Town, Xinjiang, are *F. solani* and *F. oxysporum*, consistent with previous reports [76]. However, the significantly higher isolation frequency and pathogenicity of *F*. *solani* observed in this study compared to the widely studied *F*. *oxysporum* may be related to geographical location and environmental factors [77], suggesting physiological differentiation of pathogenic fungi in different regions. Biological characterization study showed that these two pathogens had a higher growth and spore production ability at 25 °C and pH = 6, differing from the results of previous studies on the black rot pathogens of asparagus (optimum growth temperatures of 28 and 30 °C and optimum pH of 7 and 9), further confirming the ecological adaptability of the pathogens. Therefore, for the management of chili root rot, in addition to optimizing the cropping pattern, environmental factors (e.g., temperature and soil pH) can be regulated to inhibit pathogen growth, providing a new strategy for integrated disease prevention and control.

## 5. Conclusions

The present study identifies the mechanism by which chili crop rotation significantly reduces the risk of chili root rot by altering the soil microbial community structure, increasing the enrichment of beneficial bacteria, and reducing the relative abundance of pathogens. This finding provides a solid scientific basis for the use of crop rotation to improve soil health and disease resistance in chili. This study has three main limitations: (1) it compared only 10-year continuous cropping with a single-year rotation, lacking duration gradients; (2) the biocontrol functions of key strains such as *Bacillus* have not been experimentally verified; and (3) conclusions are correlative and require field validation of rotation-induced changes in root-exudate metabolites and their underlying molecular mechanisms. Future work will establish rotation duration gradients, construct synthetic consortia, and integrate metatranscriptomics and metabolomics to elucidate how root exudates recruit beneficial microbes and suppress pathogens, ultimately informing sustainable soil-health management strategies.

## Figures and Tables

**Figure 1 microorganisms-13-01806-f001:**
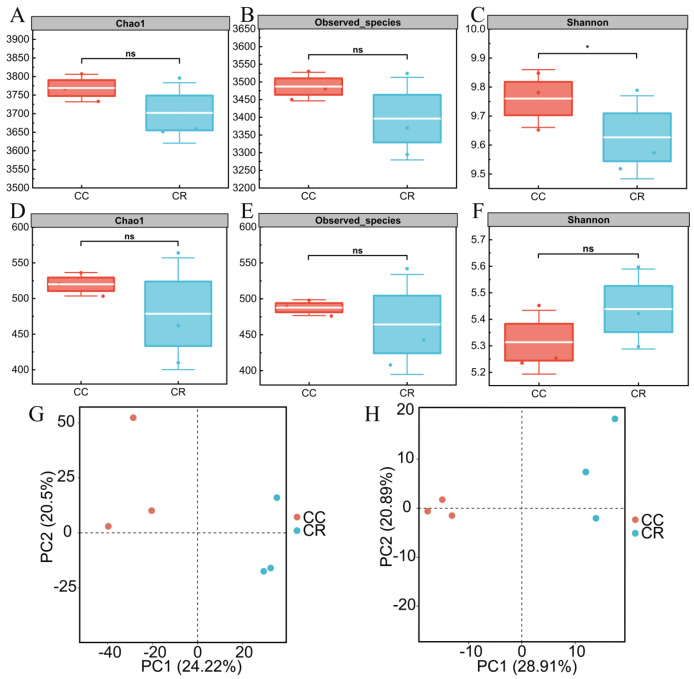
Analysis of alpha and beta diversity of rhizosphere microbial communities of chili plants with root rot under continuous (CC) and rotation (CR) cropping patterns. The box plots in the figure show the species richness-related indices of the bacterial communities in the rhizosphere soil of chili plants with root rot under CC and CR patterns: Chao1 index (**A**), Observed Species index (**B**), and community diversity index, the Shannon index (**C**). Chao1 index (**D**), Observed Species index (**E**), and Shannon index (**F**) indicating the diversity of rhizosphere soil fungal communities under the two treatments (*: *p* < 0.05; ns: *p* > 0.05). PCA based on the Bray–Curtis similarity index further revealed structural differences between the bacterial (**G**) and fungal (**H**) communities.

**Figure 2 microorganisms-13-01806-f002:**
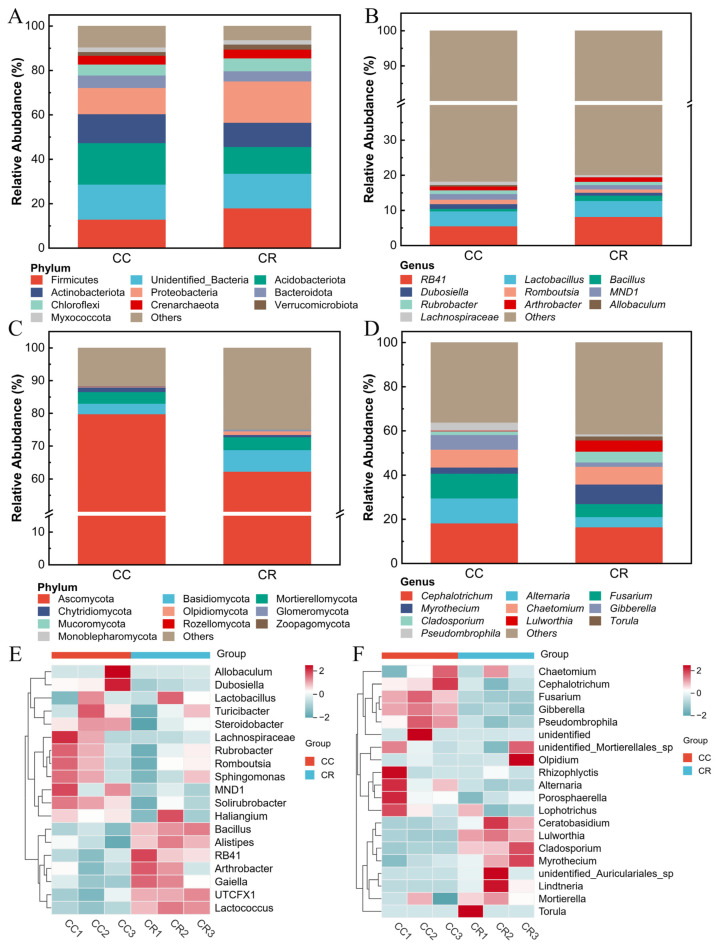
Comparison of phylum- and genus-level relative abundance and heat map of rhizosphere microbial communities of chili plants with root rot. Species composition of rhizosphere bacterial communities at the phylum (**A**) and genus levels (**B**). Species composition of rhizosphere fungal communities at the phylum (**C**) and genus levels (**D**). (**E**) Heatmap of species composition of rhizosphere soil bacteria at the genus level. (**F**) Heatmap of species composition of rhizosphere soil fungi at the genus level.

**Figure 3 microorganisms-13-01806-f003:**
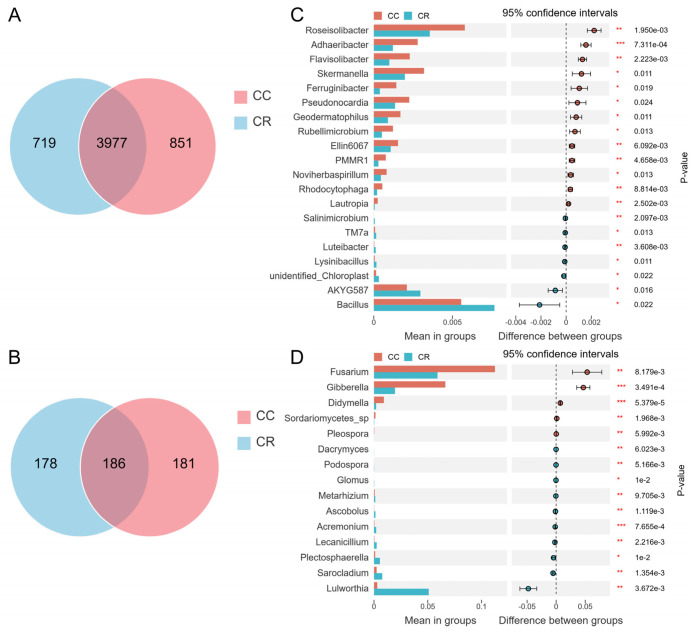
Microbial community differences based on Venn analysis and *T*-test. Venn diagram analysis of differences in the species composition of rhizosphere bacteria (**A**) and fungi (**B**). *T*-test results for microbial communities between different treatment groups at the genus level showed the distinct species and relative abundance of bacterial (**C**) and fungal (**D**) communities. Left: relative abundance of different species (bars indicate the mean values for each group); Right: confidence (circles represent 95% confidence intervals for mean differences, and colors represent groups with higher means). Significance test *p* values are shown on the far right (*: *p* < 0.05; **: *p* < 0.01, *** *p* < 0.001).

**Figure 4 microorganisms-13-01806-f004:**
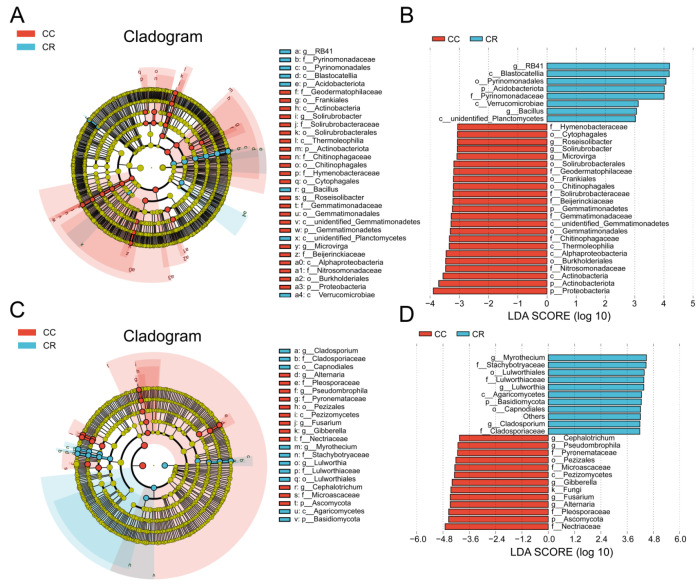
Variation in rhizosphere soil microbial communities of chili under continuous and rotational cropping patterns. (**A**) Evolutionary branching diagram of LEfSe bacterial communities with different cropping durations. Circles radiating from inside to outside represent the taxonomic level from kingdom to species, and the diameter of the small circles represents the relative abundance. Species with no significant differences are not colored, and species with significant differences are colored by biomarker following the group. (**B**) Histogram of the distribution of LDA values in the bacterial community. The bars show significantly different species with LDA scores greater than the preset value, i.e., statistically different biomarkers with an LDA value of 4; the colors of the bars represent the groups; and the lengths represent the LDA scores, i.e., the degree of influence of the significantly different species among the groups. (**C**) Branching diagram of LEfSe evolution in fungal communities with different cropping durations. (**D**) Histogram of the distribution of LDA values for fungal communities.

**Figure 5 microorganisms-13-01806-f005:**
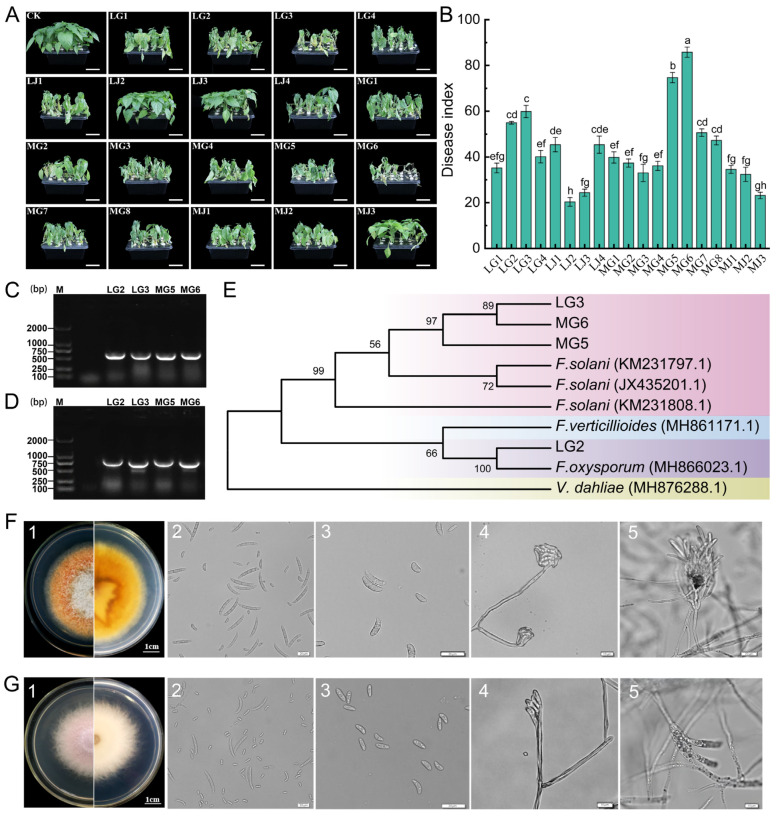
Isolation and molecular biological identification of pathogens. (**A**) Typical symptoms of chili root rot under hydroponic conditions, and the short white line in the figure represents 5 cm. (**B**) Disease index of chili root rot. Different lowercase letters indicate statistically significant differences (*p* < 0.05). (**C**,**D**) Products of PCR amplification of representative strains using fungal universal primers and specific primers, respectively, after detection in 1% agarose gel electrophoresis. (**E**) Phylogenetic tree based on ITS and EF gene sequences using the maximum likelihood method. (**F**,**G**) Morphological identification of isolates MG6 and LG2, respectively (1: colony morphology; 2: large conidium; 3: small conidium; 4: Conidial peduncle; 5: spore-producing cells).

**Figure 6 microorganisms-13-01806-f006:**
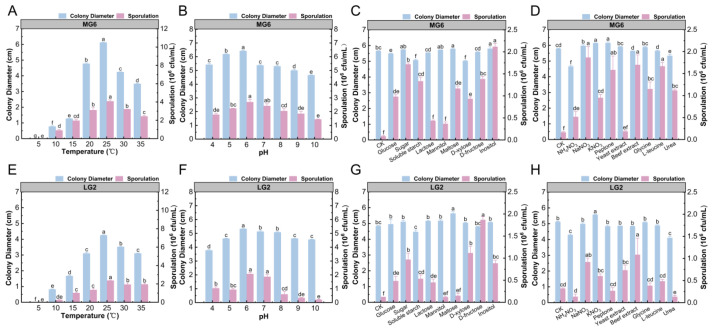
Growth and spore formation of MG6 and LG2 strains under different environmental conditions. This figure shows the changes in colony diameter (blue) and sporulation (pink) of MG6 and LG2 strains under different environmental conditions: (**A**,**E**) temperatures; (**B**,**F**) pH; and (**C**,**G**) carbon sources. (**D**,**H**) Effects of different nitrogen sources on colony growth and spore formation. Different letters indicate a statistically significant difference (*p* < 0.05).

## Data Availability

The raw data supporting the conclusions of this article will be made available by the authors on request.

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
