# Peer review of "Analysis of the Differences in Rhizosphere Microbial Communities and Pathogen Adaptability in Chili Root Rot Disease Between Continuous Cropping and Rotation Cropping Systems"

_microorganisms, 2025, doi:10.3390/microorganisms13081806_

Round 1

Reviewer 1 Report

Comments and Suggestions for Authors

Dear Authors, 

The manuscript contains interesting and useful data on chili monoculture and crop rotation. The data is well presented, and most of my comments relate to the interpretation of the results.

One of the most important things, that the research design is appropriate or not. I have a question regarding the design of the experiment. The fact that the rhizosphere of infected plants was used in both continuous cropping (CC) and crop rotation (CR) fundamentally calls into question the usefulness of the experimental results for comparing infection rates or identifying factors that indicate infection. Please provide detailed justification for this choice.

The title does not accurately reflect the content of the paper; I suggest changing it.

The abstract contains the following eight statements, which run throughout the entire manuscript: 

1_ CR treatment significantly reduced the alpha diversity indices

2_ Principal component analysis (PCA) revealed distinct compositional differences in bacterial and fungal communities between the treatments. 

3_ It increased the relative abundances of the Firmicutes, Proteobacteria, RB41, Lactobacillus and Bacillus genera in the bacterial community, while reducing the relative abundances of the Cephalotrichaceae, Alternaria and Fusarium in the fungal community.

4_ CR facilitated the enrichment of beneficial bacteria, such as Bacillus

5_ CC favored enrichment of pathogens, such as Firmicutes. In addition, Fusarium solani and Fusarium oxysporum were identified as the predominant pathogens causing chili root rot,

6_ CR enhanced soil health 

7_ reduced the root rot incidence 

8_ optimizing the structure of soil microbial communities, increasing the proportion of beneficial bacteria, and suppressing pathogens…

In addition, the thesis also deals with the isolation and characterization of pathogens. I recommend that these sections be accepted without modification.

These eight statements are developed and substantiated to varying degrees. I would like to add my comments and questions to these statements.

1_ According to generally accepted opinion, greater diversity means a better, more stable community. On the one hand, it would be worthwhile to examine more, different types of diversity in the manuscript, and on the other hand, this observation (the decreasing) should definitely be explained.

2_ I do not recommend any additions or changes. They used different methods to show that the two microbial communities are different.

3_ In addition to describing the changes, it would be necessary to at least briefly characterize what the described changes at the phylum or genus level actually mean. It does not make much sense to describe, for example, the RB41 genus without explanation, because we know nothing about its characteristics. One of the most important goals of the paper could be to explain the changes observed in the microbial community and to compare the composition of the communities with the results of other researchers.

4_ The “beneficial” word is without explanation. Many microbes in the soil can be considered useful or beneficial, but at least a reference to the justification is required. The Bacillus genus contains at least 250 species, so it matters which one is being referred to.

5_ Were the sequences of isolated fungal pathogens found in the NGS results?

6_ We do not know what integrated analysis is, and there is no evidence in the manuscript that CR improved soil health. The term “soils health” itself is not explained neither. And since the soil samples were taken from the vicinity of the infected plants, it is impossible to compare the properties of the healthy soil with that of the infected soil.

7_ Did you measure the root rot incidence in the experiment?

8_ “optimizing structure of soilmicrobial communities” To make this statement, we would need to know much more about what we mean by “optimal structure”.  This could be written about in the literature review.

Based on all this, I suggest reconsidering the interpretation of the results. And in the literature review, I recommend at least a brief explanation of the terms soil health and beneficial microbes.

Best regards, 

Author Response

Response to the reviewers’ comments

Reviewer #1:

Our response:

Thank you for taking the time to review our manuscript and provide a series of constructive comments. In response to your feedback, we have comprehensively revised the manuscript. The quality of the manuscript has been significantly enhanced through these revisions (specific amendments are detailed in the point-by-point responses to your comments below). An itemized explanation of how we addressed each of your comments and suggestions is provided as follows. For more details, please refer to our responses to your comments. In the revised manuscript, the modified content is highlighted in red font, while the purple text represents the jointly revised content.

Major comments:

Comment 1: The title does not accurately reflect the content of the paper. We have changed the title of the article.

Our response: We sincerely thank the reviewers for their valuable suggestions. In the revised manuscript, we changed the original title "Contrasting Rhizosphere Microecology and Dominant Pathogen Biology in Chili Fields with Root Rot under Continuous vs. Rotational Cropping Systems" to "Analysis of the differences in rhizosphere microbial communities and pathogen adaptability in chili root rot disease between continuous cropping and rotational cropping systems".

The specicific modification is: Analysis of the differences in rhizosphere microbial communities and pathogen adaptability in chili root rot disease between continuous cropping and rotation cropping systems. (Lines 2-4)

Comment 2: The fact that the rhizosphere of infected plants was used in both continuous cropping (CC) and crop rotation (CR) fundamentally calls into question the usefulness of the experimental results for comparing infection rates or identifying factors that indicate infection. Please provide detailed justification for this choice.

Our response: We appreciate your valuable comment regarding the use of infected plant rhizosphere soil in both the continuous cropping (CC) and crop rotation (CR) treatments. This indeed raises a valid concern about the potential confounding effects on comparing infection rates and identifying factors that influence infection. We fully understand this point and would like to provide a detailed explanation for our experimental design choice.The reason we chose to use rhizosphere soil from infected plants under both cropping systems (CC and CR) is based on several key considerations:

  1. Reflecting Real Agricultural Conditions

In practical agricultural settings, plants in both CC and CR systems are susceptible to varying levels of infection. Root rot diseases, such as those caused by Fusarium, are commonly observed in continuous cropping systems. While crop rotation can alleviate disease pressure to some extent, it does not completely eliminate infection. In real-world farming, it is unlikely that crop rotation will fully eradicate root rot. By sampling infected plants, we aim to study microbial community dynamics under realistic disease conditions. This approach enables us to generate more applicable findings for actual agricultural practice, as it reflects the reality of plant disease management under both cropping systems.

  1. Exploring Disease Progression Across Cropping Systems

By using infected plants, we are able to explore how disease progression affects microbial communities in both CC and CR systems. This is essential for understanding whether rotation can mitigate disease symptoms and regulate the microbial community in a beneficial way, even when infection is present. We hypothesize that crop rotation may regulate the microbial community by promoting beneficial microbes while suppressing pathogens, thereby mitigating root rot. Therefore, examining the microbial shifts in the rhizosphere of infected plants allows us to assess the role of crop rotation in reducing disease severity by modifying microbial composition, even in the presence of infection.

  1. Microbial Diversity and Disease Suppression

A central objective of our study is to investigate the relationship between microbial diversity and disease suppression. Numerous studies have shown that greater microbial diversity is often associated with improved disease resistance. A diverse microbial community can suppress pathogens through mechanisms such as competition, antimicrobial production, and plant resistance induction. While both cropping systems involved infected plants, we focus on the comparison of microbial community compositions in these systems, with particular emphasis on how crop rotation may enrich beneficial microbes (such as Bacillus.) and thus reduce root rot severity. Despite the presence of infection, comparing microbial community shifts across systems is crucial for understanding how rotation impacts microbial diversity and disease suppression.

  1. Comparing Microbial Community Structures

In this study, we compared the microbial community structures in infected plants across both CC and CR systems. We observed significant differences in the relative abundance of pathogens, such as Fusarium, and beneficial microbes, such as Bacillus, between the two cropping systems. Even though both systems involved infected plants, the microbial composition differed notably, with crop rotation favoring the growth of beneficial microbes while suppressing pathogens. These results strongly suggest that crop rotation not only alters the microbial community composition but also helps reduce disease severity by modulating microbial populations.

  1. Focus on Microbial Community Changes Rather Than Infection Rates:

While we did sample rhizosphere soil from infected plants, the primary focus of our research is not simply on comparing infection rates. Instead, we aim to investigate how different cropping systems influence microbial communities in the rhizosphere, particularly under disease conditions. We hypothesize that crop rotation can modulate microbial composition by promoting beneficial microbes and suppressing pathogens, thus mitigating root rot. Therefore, despite the presence of infection, comparing microbial communities between the two cropping systems remains crucial for understanding the potential of crop rotation to enhance soil health and disease resistance.

Justification for the Choice of Experimental Design:

In conclusion, while we acknowledge that using infected plant rhizosphere soil in both cropping systems could introduce some bias when comparing infection rates, we believe this choice was essential to reflect real-world agricultural conditions. By studying microbial community changes under disease conditions, we aim to gain a deeper understanding of how crop rotation can influence soil microbiomes, potentially reducing disease severity through beneficial microbial shifts. This design not only adds ecological realism but also provides scientific insights for future microbial-based disease management strategies.

We hope this detailed explanation clarifies our experimental design choice and underscores the relevance of our study to practical agricultural practices. We sincerely thank the reviewer for the constructive feedback and hope that our response satisfactorily addresses your concerns.

Comment 3: CR treatment significantly reduced the alpha diversity indices. According to generally accepted opinion, greater diversity means a better, more stable community. On the one hand, it would be worthwhile to examine more, different types of diversity in the manuscript, and on the other hand, this observation (the decreasing) should definitely be explained.

Our response: Thank you for your valuable feedback. We fully understand your concerns regarding the significant reduction in alpha diversity indices under the CR treatment. As you rightly pointed out, higher diversity typically correlates with a more stable and healthier community. However, the observed decrease in alpha diversity under CR warrants further explanation. Although alpha diversity decreased in our study, this does not necessarily indicate a decline in ecosystem function. We hypothesize that crop rotation may selectively enrich beneficial microorganisms, such as Bacillus and Lactobacillus, while suppressing pathogenic microbes, ultimately improving soil health and disease suppression. Thus, the reduction in diversity could reflect an optimized functional community rather than a negative ecological change.

Additionally, as you suggested, we will further explore different types of diversity, including alpha and beta diversity, in the revised manuscript. By incorporating these various diversity metrics, we aim to provide a more comprehensive assessment of the microbial community's structure and function. The analysis of alpha and beta diversity will help explain community-level differences and overall ecosystem health, thereby offering a broader context for the observed changes in alpha diversity.

We will integrate this analysis into the discussion, providing a more nuanced interpretation of the observed reduction in alpha diversity and its ecological implications, supported by relevant literature.

The specicific modification is: Alpha diversity analysis indicated that CR significantly reduced the Shannon index of the bacterial community, suggesting changes in community evenness and diversity. Although the Chao1 and Observed_species indices were lower, the changes were not statistically significant. This suggests that crop rotation may significantly affect the community structure by altering the richness and evenness of the bacterial community, but the extent of the impact is relatively moderate. Consistent with the changes in bacterial species abundance depicted in Figure 2 (A,B and E), the CR may have led to a more uneven distribution of species within the community by enriching certain beneficial bacteria (such as Lactobacillus and Bacillus) while inhibiting the growth of pathogenic microorganisms. This, in turn, resulted in a decrease in the overall evenness and diversity of the community. Therefore, although crop rotation did not significantly alter the species richness of the bacterial community, it significantly impacted the community's evenness and structure through the selective enrichment of specific microorganisms, leading to changes in the alpha diversity indices. (Lines 517-531)

Although the species richness of the fungal community (as indicated by Chao1 and Observed_species indices) did not show significant changes under CR, the increase in the Shannon index suggests that the internal structure of the community has changed.The heatmap analysis in Figure 2F further revealed the impact of CR on the fungal community, showing a significant reduction in the abundance of pathogenic fungi such as Fusarium and Gibberella under CR. This suggests that crop rotation may help reduce the abundance of these pathogens, thereby improving soil health and promoting crop growth. As previous studies on rotational strip intercropping have shown, crop rotation reduces the richness and diversity of bacterial communities in the peanut rhizosphere, while increasing the richness and diversity of fungal communities[54].A similar phenomenon was also observed in crop rotation experiments involving cotton and peanut.This study, employing α and β diversity index analysis, revealed that rotating peanuts led to a reduction in the α diversity of bacterial and fungal communities[55]. (Lines 532-545)

Comment 4: It increased the relative abundances of the Firmicutes, Proteobacteria, RB41, Lactobacillus and Bacillus genera in the bacterial community, while reducing the relative abundances of the Cephalotrichaceae, Alternaria and Fusarium in the fungal community.In addition to describing the changes, it would be necessary to at least briefly characterize what the described changes at the phylum or genus level actually mean. It does not make much sense to describe, for example, the RB41 genus without explanation, because we know nothing about its characteristics. One of the most important goals of the paper could be to explain the changes observed in the microbial community and to compare the composition of the communities with the results of other researchers.

Our response:Thank you for your thorough review. We fully agree with your suggestion and have added further explanation and discussion regarding the observed changes in the microbial community in the revised version of the manuscript.

The specicific modification is: RB41 is an important member of Acidobacteria and also a key participant in the soil nitrogen cycle. It regulates soil nitrogen forms and increases pH through nitrate reduction (NO₃⁻ → NH₄⁺). This alkaline environment can effectively inhibit the growth of acidophilic pathogens such as Fusarium. Studies have found that the abundance of RB41 is significantly positively correlated with soil pH (optimal pH 8.4), and can be used as a biological indicator of soil ecological health[63, 64]. (Lines 579-584)

In this study, CR significantly promoted the proliferation of RB41 by alleviating soil acidification caused by continuous cropping, indicating that the rotation mode can effectively improve the micro-environment of the root zone. Lactobacillus and Bacillus are functionally important beneficial microorganisms in rhizosphere ecosystems. Lactobacillus mediates direct antagonism against pathogenic fungi via secretion of organic acids and antifungal peptides[65]. In parallel, Bacillus exerts multifaceted biocontrol activity through biosynthesis of antimicrobial compounds including antibiotics and siderophores, competitive exclusion of pathogens, and induction of plant systemic resistance[66]. Current research substantiates that Bacillus achieves particularly robust biocontrol outcomes through dual mechanisms of antimicrobial production and host resistance induction[67, 68]. (Lines 584-595)

Comment 5: CR facilitated the enrichment of beneficial bacteria, such as Bacillus. The “beneficial” word is without explanation. Many microbes in the soil can be considered useful or beneficial, but at least a reference to the justification is required. The Bacillus genus contains at least 250 species, so it matters which one is being referred to.

Our response: We sincerely appreciate your valuable comments on the rigor of the terminology. We have clearly defined the concept of "beneficial microorganisms" in the Introduction section and cited relevant literature. Regarding the specific issue of the Bacillus genus, since this study is based on 16S rRNA sequencing and no specific strain isolation was conducted, we focused on functional groups with clear functions in the discussion and cited previous studies to support their probiotic and biocontrol potential. The relevant content has been integrated into the revised version. Your opinion has greatly enhanced the accuracy of the paper, and we are deeply grateful.

The specicific modification is: Beneficial microorganisms refer to functional microbial groups that establish mutualistic relationships with plants and enhance plant growth/stress resistance through multiple mechanisms, primarily including plant growth-promoting rhizobacteria (PGPR), endophytic bacteria (EB), and arbuscular mycorrhizal fungi (AMF)[25]. (Lines 92-95)

In parallel, Bacillus exerts multifaceted biocontrol activity through biosynthesis of antimicrobial compounds including antibiotics and siderophores, competitive exclusion of pathogens, and induction of plant systemic resistance[66]. Current research substantiates that Bacillus achieves particularly robust biocontrol outcomes through dual mechanisms of antimicrobial production and host resistance induction[67, 68]. (Lines 590-595)

Comment 6: CC favored enrichment of pathogens, such as Firmicutes. In addition, Fusarium solani and Fusarium oxysporum were identified as the predominant pathogens causing chili root rot. Were the sequences of isolated fungal pathogens found in the NGS results?

Our response: Thank you for your detailed review of our research. Regarding the inquiry about fungal pathogen sequence alignment, we would like to clarify that while our research employed 16S rRNA and ITS sequencing to examine overall microbial community structure changes and obtained fungal community data, we did not specifically conduct database alignment for known pathogenic fungi. Importantly, our genus-level relative abundance analysis revealed that Fusarium showed significantly higher relative abundance in CC compared with CR treatments (p<0.05, t-test), indirectly reflecting the dynamics of pathogenic fungi under different cultivation regimes. We fully acknowledge the importance of direct pathogen sequence alignment and plan to address this in future studies through targeted approaches such as pathogen-specific primer amplification.

Comment 7: CR enhanced soil health. We do not know what integrated analysis is, and there is no evidence in the manuscript that CR improved soil health. The term “soils health” itself is not explained neither. And since the soil samples were taken from the vicinity of the infected plants, it is impossible to compare the properties of the healthy soil with that of the infected soil.

Our response: Thank you very much for your valuable suggestions. In the revised manuscript, we have clarified the definition of “soil health” by referencing the classic framework proposed by Brugge et al.(2000), which defines soil health as the capacity of soil to sustain plant productivity, support ecological balance, and maintain environmental quality. In this study,CR improved soil pH, a key indicator of soil fertility and health. Furthermore, CR contributed to improved biological aspects of soil health by modulating microbial community structure, suppressing pathogenic Fusarium spp., and increasing the relative abundance of beneficial taxa such as Bacillus. Although our study did not include a comprehensive assessment of all physicochemical parameters, the scope and limitations of the work have been clearly acknowledged in the Discussion section.

The specicific modification is:Soil health serves as the cornerstone for sustainable agricultural development, with its essence lying in maintaining diverse beneficial microbial communities (including bacteria, fungi, and algae)[73]. These "soil guardians" support agricultural production through three key mechanisms: optimizing nutrient cycling efficiency, suppressing soil-borne pathogens, and enhancing crop yield and quality. Notably, soil microecological balance is condition dependent the synergistic effects of physicochemical properties and ecological functions determine the dynamic equilibrium between beneficial microbes and pathogens[74]. Of particular concern is that under soil degradation conditions, the pathogen-suppressing function of beneficial microbial communities is often the first to be compromised. Therefore, establishing a microbial function-based soil health monitoring system has become a crucial measure for ensuring sustainable agricultural development. (Lines 604-615)

Comment 8: Reduced the root rot incidence. Did you measure the root rot incidence in the experiment?

Our response:We are grateful to the reviewers for their valuable suggestions on the measurement of the incidence rate of root rot disease. In this study, we did not conduct statistical quantification of the specific incidence rate of root rot disease. Instead, based on the data analysis of sequencing, we inferred the disease suppression situation by analyzing the changes in the root-associated microbial community. However, our conclusion is consistent with previous studies, that is, rotation can effectively reduce the incidence rate of the disease (lines 120-124). We agree that directly measuring the incidence rate of the disease would further enhance the persuasiveness of the research. Future studies will consider including such evaluations to provide more comprehensive evidence.

Comment 9: optimizing the structure of soil microbial communities, increasing the proportion of beneficial bacteria, and suppressing pathogens. “optimizing structure of soilmicrobial communities” To make this statement, we would need to know much more about what we mean by “optimal structure”.  This could be written about in the literature review.

Our response: Thank you for pointing out this issue in your manuscript. We have added relevant content about optimizing the structure of soil microbial communities in the revised manuscript. The detailed modifications can be found in the following text.

The specicific modification is: Purposefully adjusting the structure of the rhizosphere microbial community in crops and actively introducing beneficial microorganisms can significantly reduce disease incidence and enhance the disease resistance of crops[28]. (Lines 100-102)

Related studies have demonstrated that SynComs (Synthetic Microbial Communities) constructed with different microbial proportions exhibit remarkable effects in plant disease management and improving crop disease resistance, providing new perspectives for the innovation and development of disease prevention and control strategies. Exogenous application of SynComs has been shown to significantly enhance plant disease resistance, as evidenced in the control of peanut root rot[29] and Astragalus root rot[30]. (Lines 102-108)

Reviewer 2 Report

Comments and Suggestions for Authors

Dear Authors, 

Please find my recommendation for "Contrasting Rhizosphere Microecology and Dominant Pathogen Biology in Chili Fields with Root Rot under Continuous vs. Rotational Cropping Systems"

  1. General remark: I recommend that authors use documents with line numbers. This will make the review process more focused and help authors identify precisely the issues raised by reviewers. In current form this will be challenging
  2. (abstract) Please check if are correct the statements regarding Firmicutes in case of CC and CR
  3. (abstract) Please introduce numerical results/data also not just qualitatively describe them; Please consider relevant statistics metrics (effect sizes, p-values, etc.)
  4. (introduction) The manuscript should more clearly highlight which are the current knowledge gaps related to the addressed thematic/objectives
  5. (introduction) Would be more benefic for manuscript if it will consider also the global literature view also; In my opinion a manuscript relying solely on domestic studies narrows relevance. 
  6. (methods) Check for clarity the first sentence from "Soil sample collection and processing" subsection
  7. (methods) Please mention the number of plots considered, number of samples per plot, number of replicates for samples/analysis, etc. 
  8. (methods) Provide GPS coordinated for the study location
  9. (methods) Please include information about soil properties - such information is indispensable in such studies
  10. (methods) Is not clear if health plant/rhizosphere soil were collected or not; same for control samples
  11. (methods) Please justify " rinsed under running water for 12 h." 
  12. (methods) Present normality/homogeneity checks/results when talk about T test and Wilcoxon
  13. (results) Sustain the statement "alpha diversity analysis revealed that the Chao1 index, Observed_species index, and Shannon index of the soil bacterial community were significantly reduced by CR, with the Shannon index under CR being significantly lower than that under CC." with statistical metrics (eq. p val). Please consider this kind of observation/recommendation through the whole text body where the case is
  14. (results) Please better present the pathogenicity verification/obtained results
  15. (discussions) In my opinion the manuscript present a broad overview of crop-rotation benefits but never anchor these benefits quantitatively to their obtained dataset from this research - this dilutes the relevance of the introduction section
  16. (discussions) It sounds that the manuscript equate the lower bacterial α-diversity with healthier soil. Up to my knowledge this contradicts ecological consensus, and without functional data this looks speculative
  17. (discussions) Up to my knowledge co-occurrence networks identify correlation, not interaction. For me presenting “mutualism” without edge-sign annotation is not  appropriate from science point of view
  18. (discussions) Through all discussion section there is a lack of critical/comaparative as well mechanistic interpretation of the obtained research results. This seriously weaken the manuscript scientific relevance
  19. Please point the current research study limitation also

Author Response

Response to the reviewers’ comments

Reviewer #2:

Our response:

We sincerely appreciate you for your time and effort in carefully evaluating our manuscript (Manuscript ID: microorganisms-3726492) and for providing such insightful and constructive comments. Each of your suggestions demonstrates profound expertise, which has not only significantly enhanced the academic quality of our paper but also prompted us to reflect more deeply on key aspects of the study. We have given your valuable feedback the utmost consideration and have meticulously addressed each point in the revised manuscript, with all modifications highlighted in blue text and accompanied by specific line numbers for easy reference. Your guidance has been immensely beneficial to us, and should you have any additional suggestions during further review, we would be most grateful to incorporate them.

Comments 1: General remark: I recommend that authors use documents with line numbers. This will make the review process more focused and help authors identify precisely the issues raised by reviewers. In current form this will be challenging.

Our response: We sincerely appreciate your valuable suggestion regarding the addition of line numbers. This highly professional and constructive recommendation will significantly enhance the review efficiency and manuscript standardization. We fully agree with your perspective and have carefully implemented this improvement by adding consecutive line numbers throughout the revised manuscript. Should you identify any further adjustments needed regarding the line numbering during your review, we would be most grateful for your guidance and will promptly make the necessary refinements.

Comments 2: (abstract) Please check if are correct the statements regarding Firmicutes in case of CC and CR.

Our response: Thanks for your careful review and valuable comments. Regarding the wording issue you raised, we've verified that CR treatment indeed increased the relative abundance of Firmicutes. We apologize for any confusion caused by our unclear original description. Following your suggestion, we've revised the relevant text to present the findings more clearly.

The specicific modification is: Compared with CC, CR treatment has altered the structure of the soil microbial community. In terms of bacterial communities, the relative abundance of Firmicutes increased from 12.89% to 17.97%, while the Proteobacteria increased by 6.8%. At the genus level, CR treatment significantly enriched beneficial genera such as RB41 (8.19%), Lactobacillus (4.56%), and Bacillus (1.50%) (P < 0.05). In contrast, the relative abundances of Alternaria and Fusarium in the fungal community decreased by 6.62% and 5.34% respectively (P < 0.05). (Lines 22-27)

Comments 3: (abstract) Please introduce numerical results/data also not just qualitatively describe them; Please consider relevant statistics metrics (effect sizes, p-values, etc.)

Our response: Thank you for your valuable suggestions. We have made revisions based on your advice and have included relevant numerical results in the revised version, ensuring that readers can easily locate the original data and the verification details. Once again, thank you for your meticulous guidance!

The specicific modification is: Compared with CC, CR treatment has altered the structure of the soil microbial community. In terms of bacterial communities, the relative abundance of Firmicutes increased from 12.89% to 17.97%, while the Proteobacteria increased by 6.8%. At the genus level, CR treatment significantly enriched beneficial genera such as RB41 (8.19%), Lactobacillus (4.56%), and Bacillus (1.50%) (P < 0.05). In contrast, the relative abundances of Alternaria and Fusarium in the fungal community decreased by 6.62% and 5.34% respectively (P < 0.05). (Lines 22-27)

Fusarium solani MG6 and F. oxysporum LG2 are the primary chili root-rot pathogens. Optimal growth occurs at 25°C, pH 6: after 5 d, MG6 colonies reach 6.42 ± 0.04 cm and LG2 5.33 ± 0.02 cm, peaking in sporulation (P < 0.05). In addition, there are significant differences in the utilization spectra of carbon and nitrogen sources between the two strains of fungi, suggesting their different ecological adaptability. (Lines 30-34)

Comments 4: (introduction) The manuscript should more clearly highlight which are the current knowledge gaps related to the addressed thematic/objectives

Our response: Thank you for your valuable suggestions. In the revised manuscript, we have added relevant content in the preface section, elaborating on the research gap between crop rotation and the dynamics of microbial communities.

Although existing studies have shown that crop rotation can significantly alter the structure of soil microbial communities, the specific mechanisms by which chili and different crops in rotation inhibit root rot disease remain unclear. Moreover, the specific microbial functional groups that are activated after rotation and how these microorganisms interact with soil-borne pathogens are still lacking in systematic research. Therefore, this study compared the changes in the rhizosphere microbial communities of chili under consecutive cropping and crop rotation patterns, and preliminarily explored the enrichment patterns of key beneficial microorganisms after rotation and their inhibitory mechanisms against pathogens, with the aim of providing theoretical basis for the green control of chili root rot. (Lines 140-149)

Comments 5: (introduction) Would be more benefic for manuscript if it will consider also the global literature view also; In my opinion a manuscript relying solely on domestic studies narrows relevance.

Our response: Thank you for your valuable suggestions. We fully agree with the importance of incorporating an international research perspective into this article. In response to your comments, we have added relevant literature. We believe that these revisions have significantly enhanced the academic value and international perspective of the article. All the modified parts have been highlighted for your review. If you have any further suggestions, we would be most grateful.

The specicific modification is: Beneficial microorganisms refer to functional microbial groups that establish mutualistic relationships with plants and enhance plant growth/stress resistance through multiple mechanisms, primarily including plant growth-promoting rhizobacteria (PGPR), endophytic bacteria (EB), and arbuscular mycorrhizal fungi (AMF)[25]. (Lines 92-95)

Purposefully adjusting the structure of the rhizosphere microbial community in crops and actively introducing beneficial microorganisms can significantly reduce disease incidence and enhance the disease resistance of crops[28]. (Lines 99-102)

Globally, crop rotation has long been recognized as a critical strategy in suppressing soil-borne pathogens and promoting beneficial microbial populations. In the U.S., corn–soybean rotations enhanced rhizosphere Pseudomonas and Bacillus populations, which suppressed Fusarium and Rhizoctonia diseases[37]. Similarly, In European and Mediterranean cropping systems, rotating wheat with tomato effectively suppresses Fusarium wilt. This rotation strategy enhances tomato performance by increasing the abundance of antagonistic microorganisms, optimizing soil chemical properties, and restructuring the soil ecological network[38]. Although existing studies have shown that crop rotation can significantly alter the structure of soil microbial communities, the specific mechanisms by which chili and different crops in rotation inhibit root rot disease remain unclear. Moreover, the specific microbial functional groups that are activated after rotation and how these microorganisms interact with soil-borne pathogens are still lacking in systematic research. Therefore, this study compared the changes in the rhizosphere microbial communities of chili under consecutive cropping and crop rotation patterns, and preliminarily explored the enrichment patterns of key beneficial microorganisms after rotation and their inhibitory mechanisms against pathogens, with the aim of providing theoretical basis for the green control of chili root rot. (Lines 132-149)

Comments 6: (methods) Check for clarity the first sentence from "Soil sample collection and processing" subsection

Our response: We sincerely appreciate the reviewer's attention to methodological details. We have carefully verified the geographical information of the sampling sites to confirm their accuracy and made meticulous revisions to the first sentence of the section.

The specicific modification is: In the primary chili cultivation region of Anjihai Town, Shawan City, Tacheng Prefecture, Xinjiang Uygur Autonomous Region, China. (Lines 161-162)

Comments 7: (methods) Please mention the number of plots considered, number of samples per plot, number of replicates for samples/analysis, etc.

Our response: Thank you for your valuable suggestion! We fully agree that highlighting the number of biological replicates in the Materials and Methods section helps enhance the rigor and reproducibility of the study. We have added the number of sampling plots, the number of diseased plant samples, and the number of rhizosphere soil samples in the "Soil sample collection and processing" section of the revised manuscript, and the specific modifications are as follows.

The specicific modification is: For comparative analysis, two distinct cropping systems were examined: a 10-year continuous chili cropping field (85°47'10"E, 44°37'12"N) and a cotton-chili rotation field with annual cycles (85°47'39"E, 44°37'31"N). (Lines 162-165)

For chili plants exhibiting root rot symptoms, we implemented a five-point sampling method: five diseased plants were collected at each of five sampling points per field, yielding 25 samples per field (50 total samples from both fields). All samples were properly labeled for subsequent pathogen isolation. (Lines 168-172)

At each sampling point, three rhizosphere soil samples from diseased chili were simultaneously collected. (Lines 172-173)

Comments 8: (methods) Provide GPS coordinated for the study location

Our response: Thank you for your valuable comments regarding the geographical precision. We have supplemented the complete GPS coordinates of sampling sites in the revised manuscript as suggested.

The specicific modification is: For comparative analysis, two distinct cropping systems were examined: a 10-year continuous chili cropping field (85°47'10"E, 44°37'12"N) and a cotton-chili rotation field with annual cycles (85°47'39"E, 44°37'31"N). The experimental sites were located 720 m apart with minimal elevation difference (< 3 m) and shared identical soil properties (grey desert soil, pH 8.1±0.4). From each system, plants showing characteristic disease symptoms and their rhizosphere soils were collected for subsequent pathological and microbiological characterization. (Lines 162-168)

Comments 9: (methods) Please include information about soil properties - such information is indispensable in such studies

Our response: We sincerely appreciate your valuable comments regarding the soil characteristics. In response to your concerns, we would like to clarify the following: This study selected adjacent plots (only 720 meters apart with < 3 meters elevation difference) for comparison. Both plots share the same grey desert soil type and had identical cultivation histories and management practices prior to the differentiation of cropping systems. We only measured the pH of the soil. The pH values of the CC and CR soils were 8.1 and 8.5 respectively. To address your suggestion, we commit to incorporating comprehensive measurements of soil nutrients (N/P/K) and texture in our follow-up studies. The current dataset sufficiently ensures the scientific validity of the rhizosphere microbial comparisons between the two cropping systems. We truly value your input, which has significantly enhanced the rigor of our research.

Comments 10: (methods) Is not clear if health plant/rhizosphere soil were collected or not; same for control samples

Our response: Thank you very much for your valuable suggestions! At the same time, we sincerely apologize for the confusion caused by the insufficient expression of key contents in the paper. During the sampling process, we did not take root zone samples from healthy plants. In our research, we compared and analyzed CC as the control. This indeed raises a concern worthy of attention. We fully understand this and are willing to provide detailed explanations for our experimental design choices. We chose to use the root zone soil from infected plants in both planting systems (CC and CR) based on the following key considerations:

  1. Reflecting Real Agricultural Conditions

In practical agricultural settings, plants in both CC and CR systems are susceptible to varying levels of infection. Root rot diseases, such as those caused by Fusarium, are commonly observed in continuous cropping systems. While crop rotation can alleviate disease pressure to some extent, it does not completely eliminate infection. In real-world farming, it is unlikely that crop rotation will fully eradicate root rot. By sampling infected plants, we aim to study microbial community dynamics under realistic disease conditions. This approach enables us to generate more applicable findings for actual agricultural practice, as it reflects the reality of plant disease management under both cropping systems.

  1. Exploring Disease Progression Across Cropping Systems

By using infected plants, we are able to explore how disease progression affects microbial communities in both CC and CR systems. This is essential for understanding whether rotation can mitigate disease symptoms and regulate the microbial community in a beneficial way, even when infection is present. We hypothesize that crop rotation may regulate the microbial community by promoting beneficial microbes while suppressing pathogens, thereby mitigating root rot. Therefore, examining the microbial shifts in the rhizosphere of infected plants allows us to assess the role of crop rotation in reducing disease severity by modifying microbial composition, even in the presence of infection.

  1. Microbial Diversity and Disease Suppression

A central objective of our study is to investigate the relationship between microbial diversity and disease suppression. Numerous studies have shown that greater microbial diversity is often associated with improved disease resistance. A diverse microbial community can suppress pathogens through mechanisms such as competition, antimicrobial production, and plant resistance induction. While both cropping systems involved infected plants, we focus on the comparison of microbial community compositions in these systems, with particular emphasis on how crop rotation may enrich beneficial microbes (such as Bacillus.) and thus reduce root rot severity. Despite the presence of infection, comparing microbial community shifts across systems is crucial for understanding how rotation impacts microbial diversity and disease suppression.

Comments 11: (methods) Please justify " rinsed under running water for 12 h."

Our response: Thank you for your meticulous review. We conducted a rigorous check of the experimental method and discovered that in the original manuscript, "rinsed under running water for 2 hours" was mistakenly written as "12 hours". We sincerely apologize for this error. In the process of isolating the pathogen of root rot disease, the core function of flowing water rinsing is to efficiently remove soil particles, humus, and saprophytic microorganisms adhering to the surface of the diseased roots through physical action. This treatment can significantly reduce the microbial load, creating conditions for subsequent gentle disinfection and avoiding excessive sterilization that may damage the target pathogen. Considering the developed nature of the chili root system, 2-hour rinsing can ensure the complete removal of surface residues. This method has been successfully applied to the isolation of the cotton wilt pathogen (Zhang et al., 2021). We have corrected the relevant statements and supplemented the references to enhance the accuracy and reproducibility of the methodology.

The specicific modification is: To isolate pathogens, the plant roots and stems were washed with clean water to remove impurities and then rinsed under running water for 2 h[42]. (Lines 196-197)

Comments 12: (methods) Present normality/homogeneity checks/results when talk about T test and Wilcoxon

Our response: Thank you for your expert critique regarding our statistical methodology. We fully agree that before conducting parametric tests (t-tests) and non-parametric tests (Wilcoxon tests), normality (Shapiro-Wilk test) and homogeneity of variance (Levene test) verification should be carried out. In accordance with your suggestions, we have thoroughly optimized the key validation steps, significantly enhancing the rigor of our analytical methodology. During the revision process, we focused on the following improvements: First, we supplemented the "Statistical Analysis" section with a complete description of the testing procedures; second, based on the results of the Shapiro-Wilk normality test and Levene’s test for homogeneity of variance (specific data are available in Data/Bacteria/Fungi/Assumptions_check_table), we confirmed that independent samples t-test were appropriate for intergroup comparisons; finally, we made corresponding revisions throughout the manuscript. The detailed modifications are outlined below.

The specicific modification is: Prior to parametric analysis, normality (Shapiro-Wilk test) and homogeneity of variance (Levene test) were performed. When both assumptions were met, an independent-samples t-test was used[48]. (Lines 273-275)

Comments 13: (results) Sustain the statement "alpha diversity analysis revealed that the Chao1 index, Observed_species index, and Shannon index of the soil bacterial community were significantly reduced by CR, with the Shannon index under CR being significantly lower than that under CC." with statistical metrics (eq. p val). Please consider this kind of observation/recommendation through the whole text body where the case is

Our response: Thank you very much for your valuable suggestions! At the same time, we sincerely apologize for the confusion caused by the insufficient expression of key contents in the paper. Additionally, we have verified the contents with significant differences in the analysis and marked them with (P < 0.05). It is worth noting that in the analysis of microbial relative abundance, we used a stacked bar chart to show the compositional differences between treatments, and the stacked bar chart can only display the relative abundance composition of each microbial genus, but cannot directly reflect the statistical significance differences. To effectively show the significance of genus-level differences, we also conducted a T-test in the future, and the significance level is given in the figure. We have made multiple revisions to the content of the manuscript. The revised content is as follows.

The specicific modification is: Alpha diversity analysis revealed significantly lower Shannon index in CR bacterial communities compared to CC (P < 0.05), though other indices (Chao1 and Observed species) showed no significant differences between treatments. In contrast, fungal communities exhibited no significant differences in any alpha diversity metrics (Chao1, Observed species, or Shannon indices) between CR and CC treatments, indicating minimal rotation effects on fungal diversity. (Lines 294-299)

At the phylum level, the dominant bacterial phyla included Firmicutes, Acidobacteriota, Actinobacteriota, and Proteobacteria (Fig. 2A), in which CR increased the relative abundance of Firmicutes and Proteobacteria by 5.07% and 6.80% compared to CC. (Lines 322-325)

and CR increased the relative abundance of RB41, Lactobacillus, and Bacillus compared to CC (Fig. 2B). (Lines 326-327)

Furthermore, compared with CC, CR reduced the relative abundances of the Ascomycota and Basidiomycota phyla by 17.51% and 0.76% respectively (Fig. 2C). (Lines 329-331)

Genus-level analysis (Fig. 2D) showed that Cephalotrichum, Alternaria, Fusarium, Myrothecium, and Chaetomium were the dominant genera, and CR reduced the relative abundance of Cephalotrichum, Alternaria, and Fusarium (1.80%, 6.62%, and 5.34%, respectively). (Lines 331-334)

Comments 14: (results) Please better present the pathogenicity verification/obtained results

Our response: We sincerely thank you for your valuable comments on the results section. We fully agree that the description of the experimental results in the original text needs to be presented in a more systematic and in-depth manner. Therefore, we have made the following improvements:

The specicific modification is: The results of the pathogen inoculation experiment showed that there was a significant differentiation in pathogenicity among the tested strains. The dynamic observation after inoculation revealed that the four strains, MG6, MG5, LG3, and LG2, could induce typical symptoms of chili root rot in the chili plants: initially, the lower leaves wilted and drooped (5-7 days after inoculation), then developed systemic chlorosis (8-12 days), and finally led to the complete death of the entire plant (14-20 days) (Figure 5A). (Lines 406-412)

Disease index statistics showed that the disease indices of these strains were significantly higher than those of the control group and other treatments (P < 0.05). Among them, MG6 had the strongest pathogenicity (disease index 85.80 ± 2.23), followed by MG5 (74.69 ± 2.22), LG3 (59.87 ± 2.69), and LG2 (54.93 ± 0.62) (Figure 5B, data are presented as mean ± standard error). It is noteworthy that the initial symptoms appeared 3 days earlier in the MG6 treatment group compared to LG2, and the disease progression rate was also faster. (Lines 412-418)

Comments 15: (discussions) In my opinion the manuscript present a broad overview of crop-rotation benefits but never anchor these benefits quantitatively to their obtained dataset from this research - this dilutes the relevance of the introduction section

Our response: We would like to express our gratitude to the reviewers for their valuable comments and suggestions. We fully understand your suggestion regarding the need to strengthen the correlation between the existing crop rotation benefits and the data of this study. Therefore, we have made comprehensive revisions to the discussion section, deleting the original descriptive content and focusing on conducting in-depth analysis of the observed changes in the microbial community under the CR treatment. We believe that these revisions have significantly enhanced the theoretical value of the study, and we sincerely hope that you can provide further guidance on the effectiveness of these revisions.

The specicific modification is: This study demonstrates that different farming practices significantly alter the structure and function of soil microbial communities, with particularly pronounced differences in bacterial taxa abundance and diversity. Under CR treatment, the relative abundances of two beneficial bacterial phyla, Firmicutes and Proteobacteria, increased by 5.07% (from 12.90% to 17.97%) and 6.8% (from 11.83% to 18.63%) respectively, compared to CC. Importantly, antagonistic bacteria commonly associated with Firmicutes, including Bacillus and Lactobacillus, showed significant enrichment in CR treatment. These results suggest that rotation practices may effectively enhance soil disease suppression capacity and ecosystem stability by rebuilding beneficial microbial communities[61]. At the genus level analysis, probiotic groups including RB41 (8.18%), Lactobacillus (4.26%) and Bacillus (1.50%) exhibited higher abundances in CR treatment, while potential pathogens such as Fusarium showed decreased abundance. These findings further confirm that crop rotation may selectively restructure microbial niches through rhizosphere environment modification, thereby promoting colonization and functional expression of beneficial microorganisms[62]. (Lines 564-578)

Comments 16: (discussions) It sounds that the manuscript equate the lower bacterial α-diversity with healthier soil. Up to my knowledge this contradicts ecological consensus, and without functional data this looks speculative

Our response: We sincerely thank the reviewers for their valuable suggestions. We fully understand your concern about the changes in microbial diversity. This study did observe a significant decrease in the Shannon index of the bacterial community after CR treatment (P < 0.05). In response to this important finding, we have conducted an in-depth analysis in the discussion section (Effect of crop rotation on microbial communities in rhizosphere soil): Firstly, based on previous studies, we pointed out that this might be due to the selective pressure induced by rotation; it is particularly important to note that although the diversity index decreased, by enriching beneficial bacterial groups such as Lactobacillus and Bacillus, important ecological functional compensation effects may have been produced. These supplementary analyses have made the research conclusion more comprehensive. Once again, we thank you for helping us improve the quality of the paper.

The specicific modification is: Alpha diversity analysis indicated that CR significantly reduced the Shannon index of the bacterial community, suggesting changes in community evenness and diversity. Although the Chao1 and Observed_species indices were lower, the changes were not statistically significant. This suggests that crop rotation may significantly affect the community structure by altering the richness and evenness of the bacterial community, but the extent of the impact is relatively moderate. Consistent with the changes in bacterial species abundance depicted in Figure 2 (A,B and E), the CR may have led to a more uneven distribution of species within the community by enriching certain beneficial bacteria (such as Lactobacillus and Bacillus) while inhibiting the growth of pathogenic microorganisms. This, in turn, resulted in a decrease in the overall evenness and diversity of the community. Therefore, although crop rotation did not significantly alter the species richness of the bacterial community, it significantly impacted the community's evenness and structure through the selective enrichment of specific microorganisms, leading to changes in the alpha diversity indices. (Lines 517-531)

Although the species richness of the fungal community (as indicated by Chao1 and Observed_species indices) did not show significant changes under CR, the increase in the Shannon index suggests that the internal structure of the community has changed.The heatmap analysis in Figure 2F further revealed the impact of CR on the fungal community, showing a significant reduction in the abundance of pathogenic fungi such as Fusarium and Gibberella under CR. This suggests that crop rotation may help reduce the abundance of these pathogens, thereby improving soil health and promoting crop growth. As previous studies on rotational strip intercropping have shown, crop rotation reduces the richness and diversity of bacterial communities in the peanut rhizosphere, while increasing the richness and diversity of fungal communities[54].A similar phenomenon was also observed in crop rotation experiments involving cotton and peanut.This study, employing α and β diversity index analysis, revealed that rotating peanuts led to a reduction in the α diversity of bacterial and fungal communities[55]. (Lines 532-545)

Comments 17: (discussions) Up to my knowledge co-occurrence networks identify correlation, not interaction. For me presenting “mutualism” without edge-sign annotation is not  appropriate from science point of view

Our response: We are grateful for the professional comments provided by the reviewers. We understand and agree that the co-occurrence network presented is a potential covariation relationship based on statistical correlations, rather than direct ecological interactions. It should be noted that in this study, no microbial co-occurrence network was constructed, nor were any related charts included. The expressions of "mutualistic" in the text are only used to describe the possible microbial ecological function trends, and their basis is the existing literature's understanding of the ecological functions of certain dominant groups (such as specific bacteria having the ability to antagonize pathogens). To avoid ambiguity, we have adjusted the relevant wording in the revised version, and the relevant modifications are as follows.

The specicific modification is: The persistent core taxa comprised functionally complementary microorganisms that have been linked to plant growth promotion in previous studies[76]. (Lines 620-622)

Comments 18: (discussions) Through all discussion section there is a lack of critical/comaparative as well mechanistic interpretation of the obtained research results. This seriously weaken the manuscript scientific relevance

Our response: We sincerely appreciate the reviewer's valuable comments regarding the discussion section. We fully acknowledge that a more in-depth mechanistic interpretation and comparative analysis of the research findings would further enhance the scientific value of this paper. It should be noted that, as this study is primarily based on 16S/ITS amplicon sequencing data for microbial community profiling, there are inherent limitations in elucidating the specific molecular mechanisms underlying the observed microbial community shifts. Additionally, as the first investigation into the microbiome of chili rotation systems in Xinjiang's arid region, comparable literature data for direct benchmarking remain scarce. Nevertheless, in the discussion section, we have made every effort to provide speculative functional analysis of key microbial taxa based on existing literature, while also highlighting the potential applications of these findings for sustainable chili cultivation. These preliminary insights will serve as an important reference for future, more in-depth mechanistic studies.

Comments 19: Please point the current research study limitation also

Our response: Thank you for your insightful comment regarding the study limitations. Our revised manuscript now explicitly addresses these key limitations in the Conclusion section. Specifically, we acknowledge: (1) the restricted comparison between long-term continuous cropping and short-term rotation systems, (2) the need for functional validation of observed microbial shifts, and (3) the correlative nature of current findings requiring field validation. These points have been framed as constructive directions for future research, emphasizing integration of multi-omics approaches and extended field trials. We believe these revisions significantly strengthen the manuscript's scholarly rigor.

The specicific modification is: However, several limitations remain in this study: the experimental design only compared long-term continuous cropping (10 years) with short-term crop rotation (1 year), lacking gradient analysis of different rotation durations; at the mechanistic level, while microbial community changes were observed, the biocontrol functions of key bacterial strains have not been experimentally validated through functional assays; current conclusions are primarily based on correlative data and require field trials to verify actual disease suppression efficacy, particularly regarding rotation-induced rhizosphere microenvironmental changes (e.g., metabolite profiles) and the molecular mechanisms underlying pathogen suppression remain to be elucidated. Future research should integrate multi-omics technologies, long-term field experiments, and diversified rotation pattern evaluations to systematically optimize cultivation strategies, thereby comprehensively enhancing the scientific validity and applicability of soil health management practices. (Lines 648-661)

Round 2

Reviewer 1 Report

Comments and Suggestions for Authors

Thank you for correcting the manuscript.

I recommend its acception in its current form. 

Best regards,

Author Response

Our response:

We sincerely thank you for your meticulous review and invaluable guidance. Every comment you provided in the previous round was carefully studied and thoughtfully incorporated into the revised manuscript. We are deeply honored and greatly encouraged by your recommendation to accept the paper in its present form. This recognition is not only an affirmation of our research but also a powerful incentive to maintain the highest standards of scientific rigor. We will take this as a fresh starting point, continue to advance the field, and strive to deliver even higher-quality work worthy of the trust placed in us by the journal and its reviewers. Once again, our heartfelt thanks! Should any further assistance be required, we stand ready to respond immediately.

Reviewer 2 Report

Comments and Suggestions for Authors

Dear Authors,

Thank you for considering to improve the "Contrasting Rhizosphere Microecology and Dominant Pathogen Biology in Chili Fields with Root Rot under Continuous vs. Rotational Cropping Systems" manuscript. Reading carefully the manuscript I noticed few things that should be considered. Please find them listed below:

  1. I would recommend a better/more clear formulation for lines 501-510 which could create confusion. In my opinion here the text confuses functional benefit with ecological diversity. Although the statements are actually consistent - crop rotation did reduce diversity (Shannon index), and this happened precisely because it created an uneven distribution by favoring beneficial bacteria over pathogens. In my opinion, the "contradiction" stems from presenting this selective enrichment as positive while simultaneously acknowledging it reduces evenness and overall diversity. In my view the text should clarify that crop rotation was functionally beneficial for agriculture despite (or rather because of) reducing microbial diversity through selective pressure.
  2. Similarly, the author has misinterpreted their own results by confusing community structure changes with richness changes when the evidence clearly shows the changes were limited to evenness patterns.
  3. L595-597: Please check and reconsider. It's looks that the author treats the shared species presence and stable community composition as equivalent concepts, but they are completely different. Please clarify
  4. L600: Please better explain the complementarity mechanisms
  5. L612-613: The author should be consistent with geographic specificity in case of al microbial groups
  6. L626-634: This should be reorganized for clarity. The current form is hard to be understand in my opinion
  7. General comments: There are many sentences through the manuscript that are too long and could be hard to understand by readers. Many of them could create confusion also. Please consider to reformulate/or at least to split them.

Author Response

Reviewer #2:

Our response:

Thank you for taking the time to review our manuscript (Manuscript ID: microorganisms-3726492) and provide a series of constructive comments. In response to your feedback, we have comprehensively revised the manuscript. The quality of the manuscript has been significantly enhanced through these revisions (specific amendments are detailed in the point-by-point responses to your comments below). An itemized explanation of how we addressed each of your comments and suggestions is provided as follows. For more details, please refer to our responses to your comments. In the revised manuscript, the modified content is highlighted in blue font.

Comments 1: I would recommend a better/more clear formulation for lines 501-510 which could create confusion. In my opinion here the text confuses functional benefit with ecological diversity. Although the statements are actually consistent - crop rotation did reduce diversity (Shannon index), and this happened precisely because it created an uneven distribution by favoring beneficial bacteria over pathogens. In my opinion, the "contradiction" stems from presenting this selective enrichment as positive while simultaneously acknowledging it reduces evenness and overall diversity. In my view the text should clarify that crop rotation was functionally beneficial for agriculture despite (or rather because of) reducing microbial diversity through selective pressure.

Our response: Thank you very much for your meticulous review and insightful suggestions regarding the statements in lines 501-510. The confusion you pointed out regarding "functional benefit" and "ecological diversity", as well as your insight into the apparent contradiction between "selective enrichment" and "decrease in uniformity", have enabled us to deeply recognize the logical deficiencies in the original text. We have revised the paragraph word for word based on this and clearly stated in the revised version: Crop rotation achieves its agronomic benefits by exerting selective pressure, moderately reducing Shannon uniformity, and directing the enrichment of functional bacterial communities. We believe this modification has fully addressed your concerns and we sincerely request your further review and correction.

The specicific modification is: Continuous medicinal-plant monocropping increases root-rot incidence to 3.5-fold that of newly reclaimed land[50], underscoring the need to break replant barriers for yield, soil health and the environment. Rather than indiscriminately raising microbial diversity, crop rotation exerts selective pressure that enriches beneficial taxa and suppresses pathogens, thereby reducing disease risk and stabilising yields. This targeted re-structuring of the microbiome is accompanied by improved soil nutrient status and lower pesticide dependence [51]. Consistent findings have been reported for chili production: adopting different cultivation modes after chili planting markedly decreased the relative abundance of potentially pathogenic fungi while increasing that of beneficial counterparts, further optimising microbial community structure and enhancing soil disease resistance and fertility. Similarly, alfalfa-forage rotations reassemble bacterial communities and mitigate replant disorders [52, 53].Thus, the agronomic benefit of rotation stems from its selective pressure, which lowers Shannon evenness while directionally enriching functional taxa and suppressing pathogens, thereby enhancing soil health and crop productivity. (Lines 496-510)

Comments 2: Similarly, the author has misinterpreted their own results by confusing community structure changes with richness changes when the evidence clearly shows the changes were limited to evenness patterns.

Our response: Thank you for highlighting the critical issue of our misstatement, in which we incorrectly equated community-structure shifts with changes in species richness. In response, we have (1) retitled the subsection as “Differences in Alpha/Beta diversity of soil microorganisms in the rhizosphere of chili under continuous cropping and rotation systems”; (2) removed the potentially misleading sentences beginning with “Consistent with the changes in bacterial species abundance depicted in Figure 2 (A,B and E)...” and “The heatmap analysis in Figure 2F...”; and (3) highlighted all revisions in blue throughout the revised manuscript to ensure that changes in evenness are no longer described as alterations in richness.

The specicific modification is: Differences in Alpha/Beta diversity of soil microorganisms in the rhizosphere of chili under continuous cropping and rotation systems. (Lines 511-512)

Crop rotation reshapes the chili rhizosphere microbiome, yet its primary impact lies in altering evenness rather than species richness. Following CR, the bacterial Shannon index declined significantly, whereas the Chao1 and Observed_species indices were slightly lower but not statistically different, indicating that CR restructures the community through selective enrichment of specific functional taxa without genuinely changing the total species pool. Likewise, fungal richness remained stable, while a higher Shannon index points to internal re-assembly, consistent with peanut rotation studies reporting decreased bacterial diversity and increased fungal diversity [54, 55]. (Lines 513-520)

PCA further revealed clear separation of both bacterial and fungal communities between the two cropping pattern, with bacteria being more sensitive to the shift in cultivation pattern [29]. Earlier work showed that continuous potato cropping can elevate bacterial diversity [56], whereas a tobacco-faba bean-lettuce-oilseed rape rotation lowered microbial diversity by modulating soil pH and other physicochemical properties [57]. In summary, a well-designed rotation regime safeguards soil health and boosts crop productivity by precisely modulating community evenness and the abundance of functional taxa.(Lines 521-528)

Comments 3: L595-597: Please check and reconsider. It's looks that the author treats the shared species presence and stable community composition as equivalent concepts, but they are completely different. Please clarify

Our response: We sincerely thank you for your meticulous review and expert guidance. It was only through your precise attention to lines L595–597 that we recognized a significant error in our wording. We apologize for the oversight and have immediately revised the passage word-for-word to ensure full accuracy. Your rigorous standards have not only enhanced the credibility of our study but also set a higher scholarly benchmark for us. We extend our deepest gratitude once again.

The specicific modification is: Meanwhile, CR markedly reduced the relative abundance of Fusarium, a pathogen implicated in a wide range of plant diseases[69-71]. (Lines 499-513)

Comments 4: L600: Please better explain the complementarity mechanisms

Our response: We sincerely thank the reviewer for the insightful comments. The present study demonstrates that CR markedly reshapes the rhizosphere microbiome, with the most pronounced effects observed in the bacterial community. Amplicon sequencing revealed a significant increase in the relative abundances of beneficial taxa such as RB41, Bacillus, and Lysinibacillus. Specifically, Bacillus recruits synergistic beneficial microbes and triggers systemic resistance in the host, whereas RB41 elevates rhizosphere pH via nitrate reduction. Together, these mechanisms contract the ecological niche of root-rot pathogens such as Fusarium, leading to a marked decline in their abundance and achieving a “beneficial-increase, pathogen-suppression” micro-ecological regulation. Given that the current work focuses on community structure, we have not yet dissected the underlying functional mechanisms in detail. Future studies will therefore address three priorities: (1) isolation and construction of synthetic microbial consortia (SynComs) centered on Bacillus-RB41 to validate their disease-suppressive efficacy in pot and field experiments; (2) integration of metatranscriptomics and metabolomics to elucidate how rotation-induced shifts in root exudates recruit beneficial microbes and suppress pathogens; and (3) establishment of rotation-duration gradients to evaluate the legacy effects of functional microbiota and their long-term contributions to soil health. We anticipate that these integrative efforts will clarify the molecular and ecological mechanisms underpinning rotation–microbe–disease interactions, providing a robust theoretical and practical foundation for the green, precision control of chili root rot.

Comments 5: L612-613: The author should be consistent with geographic specificity in case of al microbial groups

Our response: Thank you for highlighting the need for consistent geographic specificity. In response, we have revised the manuscript as follows:

  • Pathogenic fungi: We retained the phrase “Anjihai Town, Xinjiang” to specify that solani(MG6) and F. oxysporum(LG2) were identified as the dominant pathogens causing chili root rot in this locality.
  • Beneficial bacteria (Bacillus, RB41, Lysinibacillus): These genera are globally distributed root- and soil-associated taxa and are not restricted to Xinjiang. In the revised text they are described as “enriched in our study region” while explicitly noting their cosmopolitan distribution, ensuring uniform geographic referencing throughout the manuscript.

The specicific modification is: Importantly, antagonistic bacteria commonly associated with Firmicutes, including Bacillus and Lactobacillus, showed significant enrichment in CR treatment in our study region. (Lines 537-539)

This study confirms that the primary pathogens causing chili root rot in Anjihai Town, Xinjiang, are F. solani and F. oxysporum, consistent with previous reports[77]. (Lines 595-596)

A total of 132 pathogen strains were isolated from chili under CC and CR treatments in Anjihai Town, Xinjiang: 30 from stems (LJ) and 40 from roots (LG) under CC, and 15 from stems (MJ) and 47 from roots (MG) under CR. (Lines 397-399)

Comments 6: L626-634: This should be reorganized for clarity. The current form is hard to be understand in my opinion

Our response: Thank you for your meticulous review.We fully agree with your suggestions and have now re-organized the paragraph into three concise points that summarize the limitations, followed by three sequential steps for future work.This revision makes the logic clearer and the text more readable.

The specicific modification is:This study has three main limitations: (1) it compared only 10-year continuous cropping with a single-year rotation, lacking duration gradients; (2) the biocontrol functions of key strains such as Bacillus have not been experimentally verified; and (3) conclusions are correlative and require field validation of rotation induced changes in root-exudate metabolites and their underlying molecular mechanisms. Future work will establish rotation duration gradients, construct synthetic consortia, and integrate metatranscriptomics and metabolomics to elucidate how root exudates recruit beneficial microbes and suppress pathogens, ultimately informing sustainable soil-health management strategies. (Lines 514-522)

Comments 7: General comments: There are many sentences through the manuscript that are too long and could be hard to understand by readers. Many of them could create confusion also. Please consider to reformulate/or at least to split them.

Our response: Thank you for this helpful reminder. We have carefully reviewed the entire manuscript and revised or split any sentences that were overly long or potentially ambiguous. All changes are highlighted in the revised document to improve clarity. We sincerely appreciate your meticulous guidance.

The specicific modification is: Pathogens survive winter as mycelium or chlamydospores in the soil. They spread via rain splash, irrigation water, or infected seed, thriving under high humidity, wide day–night temperature swings, and continuous monocropping[19]. (Lines 75-77)

Despite previous evidence that crop rotation reshapes soil microbial communities, how rotation suppresses root rot is still unclear. (Lines 140-141)

Therefore, we compared chili rhizosphere microbes under continuous cropping and rotation. We then tracked how rotation enriches beneficial taxa and reduces root-rot pathogens, aiming to support green disease management. (Lines 144-146)

Rhizosphere microbes are vital for plant health and productivity[58]. They are also the frontline where soil-borne diseases develop and where microbes, pathogens, and plants interact[59, 60]. (Lines 530-532)

Although CR reshaped the microbial community, Venn analysis revealed shared OTUs across both treatments. This suggests a stable core microbiome persists in chili rhizospheres. (Lines 584-586)

Moreover, t-tests and LEfSe analyses consistently showed that CR significantly enriched the plant-growth-promoting bacteria Bacillus, Lysinibacillus, and TM7a, while reducing the relative abundances of the pathogenic fungi Fusarium and Gibberella. These results further confirm that rotation suppresses chili root rot by restructuring the microbial community. (Lines 589-593)